# *DiLA*: Disentangled Latent Action World Models

**Tianqiu Zhang** [* 1 2 3]  **Muyang Lyu** [* 1 2 3]  **Yufan Zhang** [3]  **Fang Fang** [1 3]  **Si Wu** [1 2 3]

disentangled-latent-action-world-models.github.io

## Abstract

Latent Action Models (LAMs) enable the learning of world models from unlabeled video by inferring abstract actions between consecutive frames. However, LAMs face a fundamental trade-off between action abstraction and generation fidelity. Existing methods typically circumvent this issue by using two-stage training with pre-trained world models or by limiting predictions to optical flow. In this paper, we introduce *DiLA*, a novel **Di**sentangled **L**atent **A**ction world model that aims to resolve this trade-off via content-structure disentanglement. Our key insight is that disentanglement and latent action learning are co-evolving: the predictive bottleneck inherent in latent action learning serves as a driving force for disentanglement, compelling the model to distill spatial layouts into the structure pathway while offloading visual details to a separate content pathway for generation. This synergy yields a continuous, semantically structured latent action space without compromising generative quality. *DiLA* achieves superior results in video generation quality, action transfer, visual planning, and manifold interpretability. These findings establish *DiLA* as a unified framework that simultaneously achieves high-level action abstraction and high-fidelity generation, advancing the frontier of self-supervised world model learning.

[1]Peking-Tsinghua Center for Life Sciences, Academy for Advanced Interdisciplinary Studies, IDG/McGovern Institute for Brain Research, Peking University. [2]Center of Quantitative Biology, Peking University. [3]School of Psychological and Cognitive Sciences, Key Laboratory of Machine Perception (Ministry of Education), Peking University. Correspondence to: Si Wu <siwu@pku.edu.cn>.

*Proceedings of the 43^{rd} International Conference on Machine Learning*, Seoul, South Korea. PMLR 306, 2026. Copyright 2026 by the author(s).

## 1. Introduction

World models (Friston, 2010; Sutton, 1991; Ha & Schmidhuber, 2018; Hafner et al., 2020) have emerged as a cornerstone for autonomous agents (Reed et al., 2022), enabling planning (Sobal et al., 2025), simulation (He et al., 2025), and policy learning (Hafner et al., 2020; Hansen et al., 2024; Hafner et al., 2024) in complex environments. By learning to predict future states, these models implicitly capture the underlying latent dynamics of the physical world (Garrido et al., 2025; Bar et al., 2024; Zhou et al., 2024; Assran et al., 2025). However, traditional approaches rely heavily on action-labeled datasets, which are scarce and expensive to scale compared to the vast availability of unlabeled video data (Venkataramanan et al., 2023). To bridge this gap, Latent Action Models (LAMs) (Bruce et al., 2024; Schmidt & Jiang, 2024) have been introduced to infer latent actions directly from unlabeled videos, serving as surrogates for explicit control signals (Ye et al., 2024; Chen et al., 2024b; Bu et al., 2025). A LAM consists of two core components: an Inverse Dynamics Model (IDM), which extracts latent actions from consecutive frames, and a Forward Dynamics Model (FDM), which predicts future states by conditioning past observations on these inferred actions.

Despite their promise, current LAMs face a fundamental dilemma, which we term the "**LAM Trade-off**" (Gao et al., 2025; Liu et al., 2025): the tension between action abstraction and generation quality. To ensure actions are transferable and abstract, models typically impose strong predictive bottlenecks, such as Vector Quantization (Van Den Oord et al., 2017; Bruce et al., 2024; Schmidt & Jiang, 2024; Ye et al., 2024) or variational bottleneck (Kingma & Welling, 2013; Gao et al., 2025). While these priors encourage abstraction, they often disrupt the intrinsic manifold of the latent action space, leading to over-simplification and degraded video generation fidelity (Garrido et al., 2026). Conversely, relaxing these bottlenecks improves generation but yields entangled representations where latent actions capture irrelevant visual details rather than pure dynamics (Nikulin et al., 2025).

Most existing approaches prioritize abstraction at the expense of generation quality (Schmidt & Jiang, 2024; Ye et al., 2024; Chen et al., 2024b). They often adopt a disjoint,

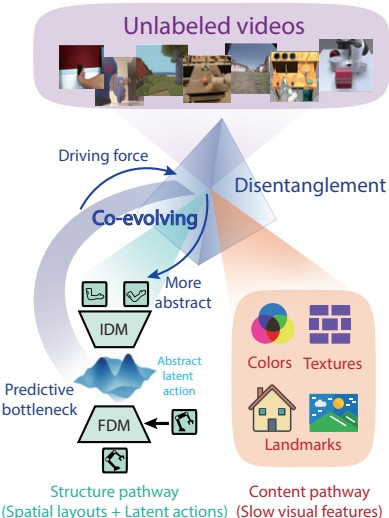

*Figure 1.* **Co-evolving of latent actions and disentanglement.** To resolve the "LAM Trade-off", *DiLA* jointly learns abstract latent actions and content-structure disentanglement. By imposing a restricted predictive bottleneck, the latent action model drives the disentanglement of spatial structures from content semantics. Conversely, this disentanglement provides structural layout inputs that facilitate the learning of highly abstract latent actions.

two-stage training paradigm, discarding the FDM in favor of a separate, pre-trained diffusion model for video generation (Bruce et al., 2024; Gao et al., 2025). To learn more abstract latent actions, some recent works extract optical flow or depth maps as inputs or supervision targets (Kim et al., 2025; Nikulin et al., 2025; Bi et al., 2025). Conceptually, these methods function as an explicit separation of motion-related spatial layouts from content details.

This observation motivates our investigation into two pivotal questions:

1. *Can disentanglement learning and latent action learning be co-optimized?*

2. *How can we simultaneously achieve abstract latent actions and high-fidelity prediction?*

In this paper, we argue that the key to resolving the "LAM Trade-off" lies in **disentanglement**. We propose *DiLA*, a novel **Di**sentangled **L**atent **A**ction world model that learns to decouple video sequences into **structure** (dynamics-relevant spatial layout) and **content** (dynamics-irrelevant visual appearance and texture). The separation is functional: features needed to model latent dynamics under the prediction bottleneck are assigned to the structure pathway. As illustrated in Fig. 1, instead of learning latent actions from entire visual features, *DiLA* forces the latent action to predict only the structural layout dynamics. This constraint facilitates the learning of content-invariant latent actions and further enforces disentanglement: to minimize structure

prediction error, the model is incentivized to distill motion dynamics into the latent action while offloading static visual details to a separate content pathway. The structure pathway captures motion-related spatial information (e.g., positions and shapes), whereas the content pathway maintains visual details for high-fidelity generation. We show that while difficult individually, the co-evolution of disentanglement and latent action learning creates a synergistic loop that significantly enhances representation learning.

To validate the capabilities of *DiLA*, we conducted experiments on a large-scale suite of datasets covering human activity, robot manipulation, and outdoor navigation. Our benchmarks demonstrate superior performance against leading methods, particularly in maintaining high generation quality while transferring actions across different embodiments. We also show that *DiLA* learns a structured, interpretable latent action space that effectively supports downstream visual planning. Collectively, these results confirm that disentangling structure from content is key to achieving both abstract action learning and high-fidelity generation.

Our contributions are summarized as follows:

- We present *DiLA*, the first disentangled latent action world model that reconciles the inherent trade-off between latent action abstraction and generation fidelity via content-structure disentanglement.

- Co-evolution of latent action and disentanglement. The predictive bottleneck acts as a driving force to isolate spatial layouts from appearance. In turn, this separation facilitates the learning of abstract latent action.

- We demonstrate *DiLA*'s capabilities through extensive experiments, covering cross-embodiment action transfer, visual planning, and the interpretable analysis of latent manifolds on out-of-distribution datasets.

## 2. Related Works

**Latent action models.** World models rely on explicit action labels to predict future states (Ha & Schmidhuber, 2018; LeCun, 2022; Assran et al., 2025). Latent Action Models (LAMs) mitigate this by inferring latent actions directly from videos (Bruce et al., 2024; Schmidt & Jiang, 2024). A challenge in LAMs is learning an abstract latent action space. Most approaches impose information bottlenecks to constrain these representations. For instance, Vector Quantization (VQ) is widely used to learn discrete latent spaces (Bruce et al., 2024; Schmidt & Jiang, 2024; Ye et al., 2024; Chen et al., 2024b;a; 2025b; Routray et al., 2025; Wang et al., 2025a), whereas other studies suggest that continuous latent spaces offer superior semantic continuity (Gao et al., 2025; Liang et al., 2025; Yang et al., 2025; Garrido et al., 2026). Beyond bottlenecks, some methods incorporate

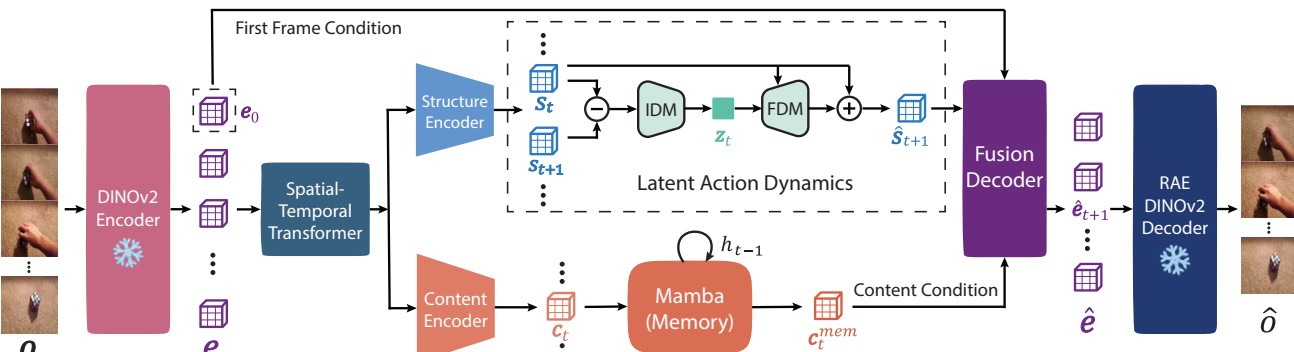

**Figure 2.** **Architecture of *DiLA*.** Video features extracted via DINOv2 and a ST-Transformer are decoupled into two pathways. The **structure pathway** learns abstract latent actions to predict the next structural state $\hat{s}_{t+1}$ under an information bottleneck constraint. The **content pathway** processes features via Mamba to maintain historical context. A fusion decoder combines the predicted structure with content and initial-frame conditioning to reconstruct the target DINOv2 embedding, visualized using a pretrained RAE decoder.

auxiliary supervision—such as sparse action labels (Nikulin et al., 2025), proprioceptive state (Chen et al., 2025b), or language instructions (Bu et al., 2025)—to encourage content invariance. Others focus on constraining input modalities using depth maps (Kim et al., 2025) or optical flow (Nikulin et al., 2025; Bi et al., 2025; Fang et al., 2025). Most works utilize latent actions primarily for VLA pre-training (Ye et al., 2024; Chen et al., 2025b). Consequently, the FDMs are discarded despite their resemblance to world models, with video generation handled by distinct, pretrained world models conditioned on learned latent actions (Bruce et al., 2024; Gao et al., 2025). While concurrent work (Garrido et al., 2026) has proposed single-stage training, it relies solely on the latent action bottleneck, missing the benefits of explicit content-structure disentanglement.

**Content and structure disentanglement.** Existing approaches like DSVAE (Yadav et al., 2023) and ContextWM (Wu et al., 2023a) attempt to split latent spaces into static and dynamic variables using ELBO-based objectives. However, these formulations frequently suffer from information leakage, resulting in entangled features. While Dyn-O (Wang et al., 2025b) improves upon this by decoupling object-centric features into dynamics-agnostic and dynamics-aware components, it relies on explicit object biases. Another related line of work exploits temporal or geometric bottlenecks to learn representations that factorize appearance from motion-related structure. For example, Rhodin et al. (Rhodin et al., 2018) learn geometry-aware representations for 3D human pose estimation by using multiview and temporal constraints, encouraging pose-related geometry to be separated from nuisance appearance factors. However, these methods typically use the bottleneck as a regularizer for pose, geometry, or motion representation. To date, no existing approach utilizes the predictive bottleneck of latent action models as the primary mechanism to drive disentanglement learning.

## 3. Method

This section details *DiLA*, a disentangled latent action world model that achieves disentanglement by processing video sequences through two specialized pathways. The first, a **structure pathway**, isolates motion-related spatial layouts to learn latent actions that are invariant to visual contents, enforced via a strict information bottleneck. The second, a **content pathway**, extracts and memorizes temporal-invariant visual features over time. Future embeddings are generated by a Fusion Decoder that recombines these structural and content representations. We employ a DINOv2 encoder (Oquab et al., 2024) for feature extraction and an RAE decoder (Zheng et al., 2025) for visualization. Specifically, *DiLA* formulates prediction entirely in the latent space (similar to JEPA (LeCun, 2022; Assran et al., 2025)), eliminating the need for pixel-level reconstruction during training. The model architecture is illustrated in Fig. 2.

To be specific, *DiLA* initiates processing by extracting visual embeddings $e_{0:t}$ from video clips $o_{0:t}$, which are then refined by a spatial-temporal Transformer (Ye et al., 2024). Spatial attention layers model global spatial dependencies, while temporal attention layers utilize causal masking to restrict information flow to historical contexts. To further enforce temporal causality, we integrate rotary position embeddings (Su et al., 2024) into the temporal attention.

**The structure pathway.** First, a structure encoder compresses tokens into structure embeddings $s_{0:t}$. The IDM then takes these embeddings to compute abstract latent actions $z_{0:t-1}$. In self-supervised settings, the temporal difference of structure embeddings ($\Delta s_{0:t-1}$) effectively represents dominant motion changes. Accordingly, the IDM processes these differences using 3D convolutional blocks: spatial kernels capture translation-invariant global features, while temporal kernels aggregate bidirectional context to form time-dependent latent actions ($d_z = 256$). The FDM is designed as a lightweight spatial-temporal Transformer. It

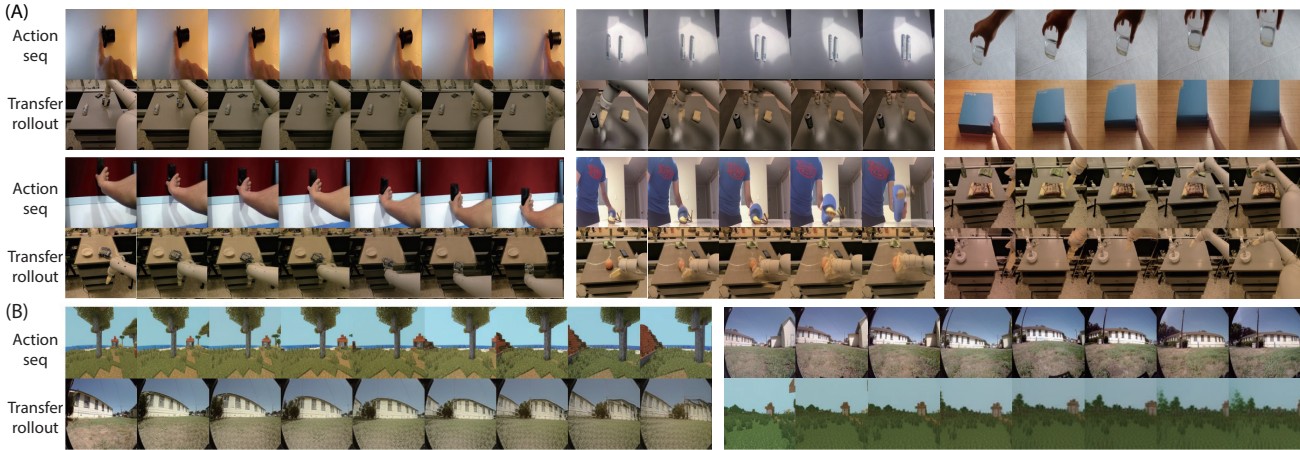

*Figure 3.* **Action transfer.** (A) **Cross-embodiment and intra-domain transfer**. Left: Human-to-robot latent action transfer. Middle: Semantic transfer across diverse objects and viewpoints. Right: Intra-domain transfer (human-to-human and robot-to-robot). (B) **Navigation transfer**. Action transfer between virtual simulations and real-world navigation environments.

generates the next state by predicting displacement vectors based on the current state $s_{0:t-1}$, conditioned on the latent actions $z_{0:t-1}$ using AdaLN-zero (Peebles & Xie, 2023). The final prediction is formulated as a residual update:

$$z_{0:t-1} = \mathrm{IDM}(\Delta s_{0:t-1})$$
$$\hat{s}_{1:t} = s_{0:t-1} + \mathrm{FDM}(s_{0:t-1}, z_{0:t-1}). \quad (1)$$

Crucially, the temporal difference $\Delta s_{0:t-1}$ serves a dual purpose: it acts as an information bottleneck to enforce abstraction and implicitly functions as the driving force for disentanglement. Since this pathway is tasked with predicting the structure embedding $s$ (rather than the full visual embedding $e$) using these abstract latent actions, a dual optimization pressure emerges. To minimize prediction error, the model is incentivized to: (1) distill only abstract dynamics into the latent action, and simultaneously (2) ensure the target $s$ retains only dynamics-correlated spatial layouts. This mutual adaptation renders the prediction task tractable. Consequently, the model naturally purges content details from $s$, as such high-entropy information cannot be effectively compressed into the low-dimensional latent action, which would otherwise impede accurate prediction.

**The content pathway.** A content encoder is used to compress tokens into embeddings $c_{0:t}$. To aggregate historical content information, we utilize the Mamba architecture (Gu & Dao, 2024). Unlike traditional RNNs, Mamba offers superior training parallelization and stable memory retention over long sequences. Functionally, the Mamba module mimics the principle of Slow Feature Analysis (Wiskott & Sejnowski, 2002): it aggregates static features as they are revealed, differing from the structure pathway's focus on dynamics. In a POMDP, content is static in the world state but dynamic in the belief state due to partial observability. We offload these belief updates to the content pathway, thereby allowing the structure pathway to

specialize strictly in modeling physical dynamics. Furthermore, this long-term memory allows the model to preserve information about temporally occluded backgrounds, as well as to infer unobserved regions in new scenes. The output of the memory module at time $t$ is formulated as: $c_t^{\mathrm{mem}} = \mathrm{Mamba}(c_t, h_{t-1})$.

**Fusing content and structure.** We employ a spatial-attention Transformer equipped with a dual cross-attention mechanism as the Fusion Decoder: with $\hat{s}$ acting as the queries, the first cross-attention module utilizes the content memory $c^{\mathrm{mem}}$ as keys and values, followed by a second that attends to the initial visual embedding $e_0$ as keys and values. Conditioning on $e_0$ is crucial, as it supplies high-frequency details lost during compression. The decoding process at time $t$ is:

$$\mathrm{Dec}_\theta(\hat{s}_{t+1}, e_0, c_t^{\mathrm{mem}}) = \hat{e}_{t+1}. \quad (2)$$

**Latent rollouts.** Unlike prior approaches (Gao et al., 2025; Routray et al., 2025) that perform rollouts in high-dimensional observation space, *DiLA* generates rollouts directly within the latent structure space via autoregressive iteration. At time step $t$, the model predicts the subsequent structure state by applying the FDM to the previously predicted state $\hat{s}_t$ and the current latent action $z_t$:

$$\hat{s}_{t+1} = \hat{s}_t + \mathrm{FDM}(\hat{s}_t, z_t). \quad (3)$$

Upon obtaining $\hat{s}_{t+1}$, we reconstruct the visual embedding $\hat{e}_{t+1}$ via the Fusion Decoder (Eq. 2). Subsequently, the content memory is updated to $\hat{c}_{t+1}^{mem}$ with this newly generated $\hat{e}_{t+1}$ to condition the next step of the rollout.

**Training objectives.** We train *DiLA* using a self-supervised *teacher-forcing* paradigm that relies solely on video sequences, thereby eliminating the need for ground-truth action labels. Throughout training, both the DINOv2

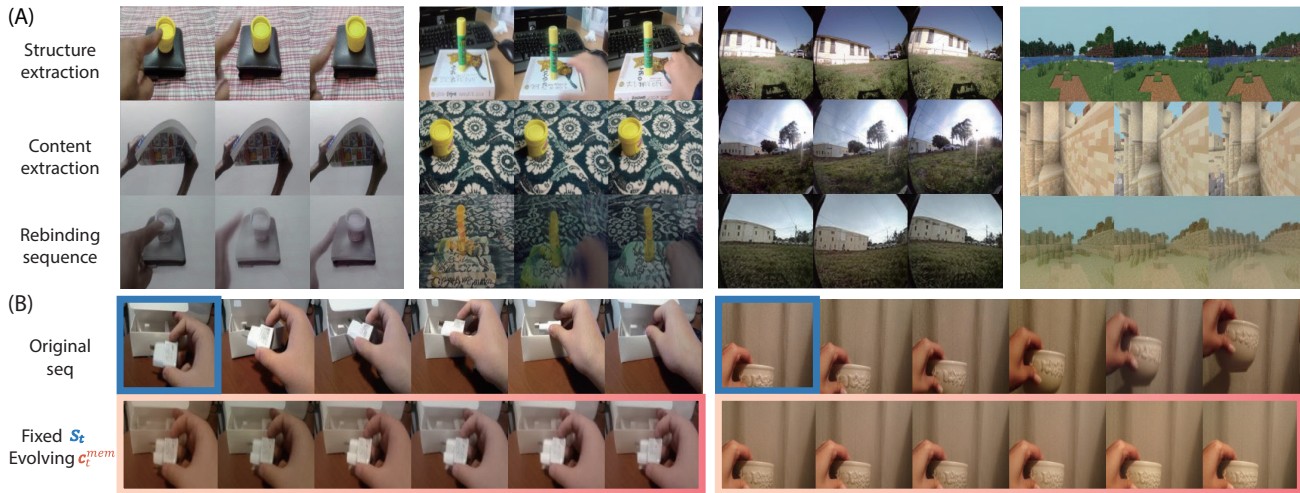

*Figure 4.* **Content and structure disentanglement.** (A) **Rebinding**: Structure from a source sequence is fused with content from a reference sequence. The output retains the source's spatial dynamics and the reference's appearance. (B) **Motion Isolation**: Fixing the structure embedding $s$ results in a static sequence, confirming that content memory $c^{\text{mem}}$ encodes no motion information.

encoder and the RAE decoder remain frozen. The total training objective is a weighted combination of visual latent prediction, structure prediction, latent action consistency, and regularization losses:

$$
\begin{aligned}
\mathcal{L}_{\text{total}} &= \lambda_{e}\mathcal{L}_{e} + \lambda_{s}\mathcal{L}_{s} + \lambda_{z}\mathcal{L}_{z} + \lambda_{\text{reg}}\mathcal{L}_{\text{reg}} \\
\mathcal{L}_{e} &= \|e_t - \hat{e}_t\|_2 \\
\mathcal{L}_{s} &= \|\Delta s_t - \text{FDM}(s_t, z_t)\|_2 \\
\mathcal{L}_{z} &= \|\text{IDM}(s_{t+1} - s_t) - \text{IDM}(\hat{s}_{t+1} - s_t)\|_2 \quad (4) \\
\mathcal{L}_{\text{reg}} &= \|z_t\|_2 + \frac{\sum_t m_t \cdot (\cos(z_t^{\text{fwd}}, z_t^{\text{bwd}}) + 1)^2}{\sum_t m_t + \epsilon} \\
&\quad + 0.5 \cdot \exp(-10 \cdot \sigma_z),
\end{aligned}
$$

where $z_t^{\text{fwd}} = \text{IDM}(s_{t+1} - s_t)$ and $z_t^{\text{bwd}} = \text{IDM}(s_t - s_{t+1})$ represent the forward and backward latent actions, respectively. The mask $m_t = \mathbb{I}[\|s_{t+1} - s_t\| > \tau]$ filters out static frames. Adopting the principle of group symmetry (Koyama et al., 2023; Hayashi et al., 2025), we employ a cosine similarity objective in $\mathcal{L}_{\text{reg}}$ to enforce that inverse temporal transitions yield opposite latent action vectors. This geometric constraint compels the latent space to align with meaningful motion dynamics while suppressing stochastic, irrelevant distractors. To further constrain this manifold, we introduce additional regularization terms targeting the norm and variance of the latent actions. We constrain the vector norm to maintain a compact manifold and prevent topological distortion, while regularizing the variance to maximize information entropy.

## 4. Experiments

In this section, we evaluate the performance and properties of *DiLA* through a series of experiments. We first validate the abstraction and transferability of the learned latent

actions via cross-domain action transfer tasks (Sec. 4.1). Next, we benchmark *DiLA* against state-of-the-art baselines to assess generation quality (Sec. 4.2). We then investigate the mechanism of disentanglement, using rebinding experiments to demonstrate the model's ability to flexibly decouple structure from content (Sec. 4.3). Ablation studies further reveal a synergistic relationship between disentanglement and latent action learning, showing that both are essential for optimal performance (Sec. 4.4). We also probe the interpretability of the latent action space, mapping low-dimensional manifolds to action semantics (Sec. 4.5). Finally, we assess the model's utility in downstream visual planning tasks (Sec. 4.6).

*DiLA* is trained for 30k + 1k iterations at a batch size of 32 with 16-frame clips across a diverse corpus of video datasets, including Something-Something-v2 (SSv2) (Goyal et al., 2017), RT-1 (Brohan et al., 2022), RECON (Shah et al., 2021), and LoopNav (Lian et al., 2025). This extensive dataset covers a wide spectrum of physical scenarios. The initial 30k iterations are conducted under a *teacher-forcing* regime. Subsequently, we fine-tune the model for an additional 1k iterations using a latent rollout paradigm (Eq. 3) to improve temporal stability. Further implementation details are provided in Appendix A.

### 4.1. Action Transfer

The action transfer task evaluates two core capabilities: (1) the abstraction and content-invariance of the latent actions, which must isolate motion dynamics to facilitate cross-domain transfer; and (2) the generative fidelity of the world model. Specifically, the model is tasked with synthesizing physically consistent sequences from an initial frame and a sequence of transferred latent actions. Leveraging

DINOv2 embeddings for fine-grained object segmentation capability, *DiLA* effectively learns latent action extraction and object-level transfer (Fig. 3).

*DiLA* successfully extracts latent actions from human activities and applies them to robotic arms, achieving cross-embodiment action transfer. Beyond simple translations (e.g., mapping a moving hand to a moving robot end-effector), *DiLA* enables complex semantic transfer—such as "picking up" an object—even across significant viewpoint changes. Furthermore, the content memory module plays a critical role by generating plausible predictions for regions unobservable in the initial frame, ensuring visual consistency. We also demonstrate success in intra-domain transfer and navigation transfer, where egocentric camera movements are effectively transferred between virtual and real-world environments. Collectively, these results confirm that *DiLA* not only captures abstract latent actions but also generates physically plausible sequences.

To quantitatively measure the quality of latent action transfer, we introduce the action cycle transfer metric (Garrido et al., 2026) in Section 4.4, which assesses whether the learned latent actions are sufficiently abstract to support transfer across different contexts. Corresponding results are reported in Table 2. Specifically, latent actions are first inferred from a source video, then transferred to a target video. From the rollout of the target video using these transferred actions, the actions are re-inferred and applied back to the source. If the semantics of the transferred actions are preserved, the resulting increase in prediction error should remain small. This provides a practical way to evaluate cross-embodiment transfer without requiring ground-truth transferred target videos. Using this metric, we mainly compared *DiLA* against ablated variants on SSv2 and RT-1. Given the vast diversity in egocentric locomotion and scene appearance, these results offer rigorous quantitative evidence of robust transfer across different scenarios, going far beyond mere qualitative similarity.

## 4.2. Video Generation Quality

We benchmark *DiLA* against several state-of-the-art methods, including LAPA (Ye et al., 2024), Moto (Chen et al., 2024b), AdaWorld (Gao et al., 2025), and villa-X (Chen et al., 2025b). Since AdaWorld relies on an external pre-trained diffusion model, we also evaluate its standalone LAM component for fairness. To evaluate generation fidelity, we employ an autoregressive rollout of 16 frames on the SSv2 and RT-1 datasets. We utilize two standard metrics: SSIM (Wang et al., 2004) for structural consistency and LPIPS (Zhang et al., 2018) for perceptual similarity. As detailed in Table 1, *DiLA* outperforms the majority of baselines across all metrics. To isolate the influence of the RAE decoder on generation quality, we examine the performance

of the ablated *DiLA* w/o content model. This comparison not only validates the effectiveness of our content-structure disentanglement but also underscores the role of the content pathway in sustaining high-fidelity video generation.

*Table 1.* **Baselines comparison on video generation.**

| MODEL | SSv2 | | RT-1 | |
|---|---|---|---|---|
| | SSIM ↑ | LPIPS ↓ | SSIM ↑ | LPIPS ↓ |
| LAPA | $0.637 \pm 0.035$ | $0.565 \pm 0.021$ | $0.491 \pm 0.014$ | $0.595 \pm 0.005$ |
| MOTO | $0.555 \pm 0.043$ | $0.593 \pm 0.022$ | $0.762 \pm 0.023$ | $0.284 \pm 0.016$ |
| ADAWORLD(FDM) | $0.625 \pm 0.029$ | $0.576 \pm 0.016$ | $0.554 \pm 0.014$ | $0.549 \pm 0.012$ |
| ADAWORLD | $\mathbf{0.674 \pm 0.008}$ | $0.521 \pm 0.022$ | $0.634 \pm 0.013$ | $0.429 \pm 0.009$ |
| VILLA-X | $0.636 \pm 0.036$ | $0.515 \pm 0.026$ | $0.576 \pm 0.023$ | $0.477 \pm 0.021$ |
| *DiLA* W/O CONTENT | $0.594 \pm 0.050$ | $0.450 \pm 0.031$ | $0.647 \pm 0.022$ | $0.258 \pm 0.015$ |
| *DiLA* | $0.660 \pm 0.037$ | $\mathbf{0.356 \pm 0.027}$ | $\mathbf{0.774 \pm 0.010}$ | $\mathbf{0.206 \pm 0.013}$ |

## 4.3. Content and Structure Disentanglement

*DiLA* achieves the implicit disentanglement of content and structure by learning latent actions to predict future structure embeddings. To validate this, we perform a rebinding experiment, synthesizing new sequences by combining structure and content from distinct videos. Specifically, we extract structure embeddings $s_{0:t}^i$ from sequence $i$ and content memory $c_{0:t}^{\mathrm{mem},j}$ from sequence $j$. These are fused to generate a visual sequence: $\hat{e}_{1:t} = \mathrm{Dec}_\theta(s_{1:t}^i, e_0^j, c_{0:t-1}^{\mathrm{mem},j})$ where $i \neq j$. As shown in Fig. 4(A), the rebinding sequence preserves the spatial layouts of the structure source ($i$) while inheriting the appearance attributes (colors, textures, and landmarks) of the content source ($j$). In this context, the Fusion Decoder functions analogously to a style transfer generator (Huang & Belongie, 2017; Zhu et al., 2017).

To rigorously rule out the possibility of motion leakage into the content pathway, we conduct a control experiment where the structure embedding is frozen at the initial state $s_0$, while the content memory $c_{0:t}^{\mathrm{mem}}$ evolves naturally over time. We generate the sequence via $\hat{e}_{1:t} = \mathrm{Dec}_\theta(s_0, e_0, c_{0:t-1}^{\mathrm{mem}})$. Fig. 4(B) demonstrates that the resulting sequence remains completely static. This confirms that although the content memory updates over time, it encodes only temporally invariant features, thereby validating the effectiveness of our disentanglement strategy.

## 4.4. Ablation Study

**Disentanglement fails without IDM + FDM.** We conduct an ablation study to verify the hypothesis that latent action learning serves as the primary driving force for disentanglement. By removing the IDM and FDM, we create a variant (*DiLA* w/o IDM+FDM) where the structure pathway functions solely as a compressor, encoding visual tokens directly into structure embeddings $s_{0:t}$. During training, $s_{t+1}$ is passed directly to the Fusion Decoder to generate the next visual embedding $\hat{e}_{t+1} = \mathrm{Dec}_\theta(s_{t+1}, e_0, c_t^{\mathrm{mem}})$, thus bypassing latent action learning. To evaluate structural purity, we visualize reconstructions using structure embeddings alone (i.e., $\hat{e}_{0:t} = \mathrm{Dec}_\theta(s_{0:t}, 0, 0)$). As shown in Fig. 5(A), the ablated model encodes redundant content details within

the structure embeddings, whereas the full *DiLA* model retains only motion-related spatial layouts. Furthermore, rebinding experiments (Fig. 5(B)) reveal that the ablated model suffers from texture leakage, effectively inheriting appearance from the structure sequence. These results prove that the predictive bottleneck imposed by latent action learning is essential for driving the disentanglement of content and structure.

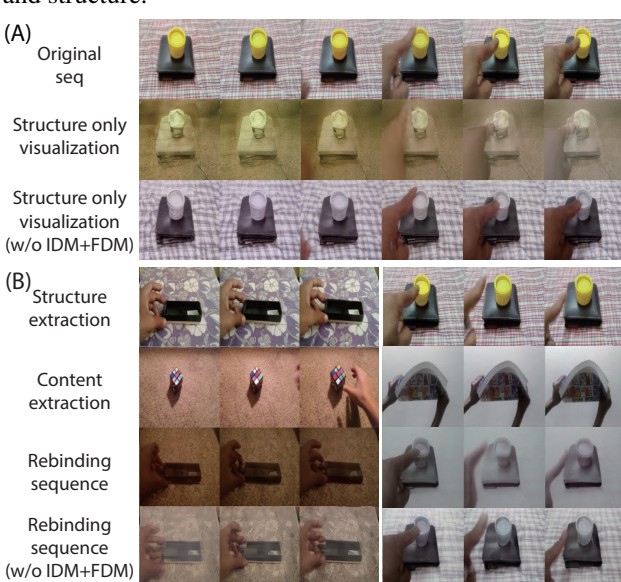

(A) Original seq

Structure only visualization

Structure only visualization (w/o IDM+FDM)

(B) Structure extraction

Content extraction

Rebinding sequence

Rebinding sequence (w/o IDM+FDM)

*Figure 5.* **Ablations on disentanglement learning.** (A) The structure embedding $s$ in *DiLA* captures motion-specific spatial layouts, whereas the ablated model (without latent action learning) retains redundant content details in $s$, resulting in poor separation. (B) In rebinding experiments, the ablated model generates artifacts where texture leaks from the structure sequence, confirming that the latent action learning is critical for preventing content leakage.

**Trade-off reappears without disentanglement.** We also conduct ablation studies to validate the hypothesis that disentanglement is crucial for learning a more accurate latent action world model. By removing the content pathway and retaining only the structure pathway, we create a variant (*DiLA* w/o content) that mirrors the standard design of current LAMs. To investigate the trade-off where higher abstraction typically compromises generation quality, we evaluate this variant on three criteria: (1) generation quality of rollouts, (2) action cycle transfer quality, and (3) convergence speed. To ensure a fair comparison of model capacity, we increase the per-patch structure embedding dimension from 32 to 128 in the ablated model. This gives the w/o content model 646.17M trainable parameters, compared with 123M for DiLA. As a result, the gap between the two models cannot be attributed to DiLA having higher capacity. We adopt the action cycle transfer metric from Garrido et al. (2026): actions inferred from a source video are transferred to a target video, then re-inferred and applied back to the source. A minimal increase in prediction error indicates that actions are robustly transferred and preserved.

As shown in Table 2, the ablated model transfers latent actions more fragilely, with LPIPS rising from 0.344 to 0.451 versus *DiLA*'s increase from 0.263 to 0.343. Its generation quality is severely compromised compared to the full *DiLA* model. Furthermore, the convergence speed is notably slower. These results confirm that the trade-off between abstraction and generation reappears in the absence of the content pathway, demonstrating that disentanglement effectively resolves this tension.

*Table 2.* **Ablations on latent action learning.** Model variants include: (1) *DiLA* w/o content, which ablates the content pathway; (2) Discrete $z$, using an NSVQ bottleneck; and (3) Gaussian $z$, using a Gaussian prior. All metrics are evaluated over 16 generation steps using LPIPS. MSE values after 10k iterations quantify convergence speed. The results are evaluated on a mixture of SSv2 and RT-1.

| MODEL | ROLLOUTS ↓ | CYCLE TRANSFER ↓ | MSE ↓ |
|---|---|---|---|
| *DiLA* W/O CONTENT | $0.344 \pm 0.030$ | $0.451 \pm 0.018$ | $0.249 \pm 0.035$ |
| DISCRETE $z$ | $0.334 \pm 0.020$ | $0.442 \pm 0.028$ | $0.262 \pm 0.033$ |
| GAUSSIAN $z$ | $0.346 \pm 0.019$ | $0.434 \pm 0.018$ | $0.265 \pm 0.024$ |
| *DiLA* | $\mathbf{0.263 \pm 0.027}$ | $\mathbf{0.343 \pm 0.022}$ | $\mathbf{0.216 \pm 0.031}$ |

**Information bottlenecks on latent actions.** Beyond the temporal difference and symmetry regularization employed in *DiLA*, we investigated alternative information bottlenecks commonly used in LAMs: vector quantization (VQ) and variational KL loss. For the discrete VQ setting, we adopted NSVQ with a codebook size of 8 and a quantized dimension of 32 (total $d_z = 512$) (Ye et al., 2024), ensuring a latent capacity comparable to *DiLA*. For the continuous variational setting, we utilized the $\beta$-VAE formulation from Gao et al. (2025) with $\beta = 10^{-4}$ and $d_z = 256$. As shown in Table 2, while both variants successfully learn abstract latent actions, they exhibit inferior generation quality and slower convergence compared to *DiLA*. We attribute this performance gap to the nature of the priors: while helpful for abstraction, the rigid priors imposed by VQ and VAE can be excessively strong, disrupting the intrinsic low-dimensional manifold of the latent action space. This distortion ultimately hampers generative fidelity and training stability.

Beyond assessing generation fidelity, we investigate the impact of information bottlenecks on the structure of the latent action space. We evaluate the quality of the learned representations via linear probing on four out-of-distribution (OOD) benchmarks unseen during training: Franka Kitchen (Gupta et al., 2019), Block Pushing (Florence et al., 2022), Push-T (Chi et al., 2023), and LIBERO Goal (Liu et al., 2023). As detailed in Table 3, *DiLA* achieves the lowest probing Mean Squared Error (MSE) across all datasets, indicating that our latent actions align more accurately with the ground truth continuous control signals. This suggests that alternative bottleneck designs are often overly restrictive, preventing the learning of fine-grained, transferable actions.

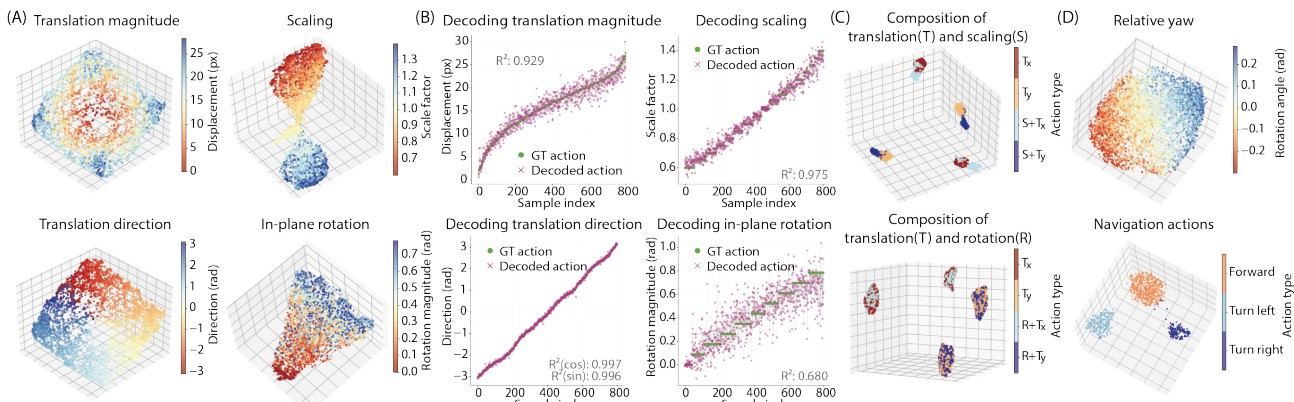

*Figure 6.* **Latent action analysis.** (A) UMAP visualization of latent actions corresponding to translation, scaling, and in-plane rotation, each forming a distinct continuous manifold. (B) Quantitative decoding validates the latent space as a meaningful low-dimensional manifold of continuous actions. (C) UMAP visualization of compositional actions merging different transformation types. (D) UMAP visualization of latent action space in navigation tasks: the model learns a continuous spectrum of relative yaw in the RECON dataset (Upper) and discrete clusters for forward and turning motions in the LoopNav dataset (Lower).

## 4.5. Latent Action Analysis

To gain deeper insight into *DiLA*'s internal representations, we analyze the learned latent action space using the *OmniObject3D* dataset (Wu et al., 2023b). Unlike the action spaces of SSv2 and RT-1, *OmniObject3D* serves as a controlled, out-of-distribution (OOD) benchmark, featuring objects of varying shapes undergoing primitive affine transformations (translation, scaling, and rotation) with customizable parameters. This control allows us to assess the semantic continuity of the latent space rigorously.

We first investigate whether *DiLA* captures the low-dimensional manifold of each transformation type. We generate sequences using a single type of transformation with randomly sampled parameters: translation (arbitrary direction/magnitude), scaling ($0.6\times$ to $1.4\times$), and in-plane rotation (0 to $\pi/4$). UMAP projections (McInnes et al., 2018) of the extracted latent actions are shown in Fig. 6(A). The translation space forms a 2D plane where small displacements cluster near the origin (red) and large displacements span the distal corners (blue), revealing a manifold structure topologically isomorphic to the physical motion. Similarly, the scaling space exhibits symmetry around the identity (no-scaling) point, while rotation actions form a continuous spectrum of rotation magnitude. Quantitative decoding of ground-truth actions (Fig. 6(B)) yields high accuracy, corroborating the visual analysis. These results confirm that *DiLA* extracts pure transition dynamics that are semantically continuous and invariant to object identity, position, scale, and orientation.

We further probe the structure of compositional actions. In the Translation+Scaling setting (Fig. 6(C)), composite actions form distinct clusters flanking the pure translation manifold. Notably, these clusters retain the topology of the scaling manifold, indicating that the latent space supports semantic compositionality. However, for Translation+Rotation, the manifolds overlap. We attribute this to the dominance of translations in the visual signal, which overshadows the in-place rotational cues in the projection. We further visualize the sequence generated by the compositional latent actions in Appendix Fig. 10.

Finally, we extend this analysis to navigation tasks (Fig. 6(D)). In the RECON dataset, *DiLA* learns a continuous spectrum of relative yaw, whereas in LoopNav, it identifies discrete clusters corresponding to forward and turning motions. This structured semantic manifold explains the model's robust performance in the action transfer tasks.

## 4.6. Visual Planning using Model Predictive Control

To validate *DiLA*'s efficacy in robotic control, we evaluate its visual planning performance on the VP$^2$ benchmark (Tian et al., 2023). We first directly use the pretrained DiLA to extract latent actions from the downstream robotic datasets. To bridge the domain gap, we train a lightweight action MLP to project ground-truth action labels into *DiLA*'s latent action space. Substituting the original IDM with this learned action MLP, we fine-tune the rest of the model on the RoboDesk and Robosuite datasets, adhering strictly to the protocol established in Gao et al. (2025).

At this stage, *DiLA* is adapted into an action-conditioned world model for MPC, but no latent action learning is performed during fine-tuning. The fine-tuned *DiLA* then serves as the dynamics model for Model Predictive Control (MPC), implemented via the sampling-based Model Predictive Path Integral (MPPI) algorithm (Williams et al., 2016). Comparative results in Table. 4 demonstrate that *DiLA* outperforms the baseline *AdaWorld* on the majority of tasks, with particularly significant gains in the "Push Button" task. These results confirm that *DiLA* is capable of surpassing pretrained

*Table 3.* **Linear probing MSE↓ across four out-of-distribution robotic benchmarks.**

| METHOD | FRANKA KITCHEN | BLOCK PUSHING | PUSH-T | LIBERO GOAL |
|---|---|---|---|---|
| DISCRETE $z$ | $0.098 \pm 0.014$ | $0.061 \pm 0.015$ | $0.023 \pm 0.004$ | $0.160 \pm 0.023$ |
| GAUSSIAN $z$ | $0.125 \pm 0.020$ | $0.102 \pm 0.023$ | $0.041 \pm 0.006$ | $0.190 \pm 0.022$ |
| *DiLA* | $\mathbf{0.073 \pm 0.014}$ | $\mathbf{0.037 \pm 0.013}$ | $\mathbf{0.009 \pm 0.003}$ | $\mathbf{0.119 \pm 0.018}$ |

*Table 4.* **Visual planning success rate on the VP$^2$ benchmark.** Results represent the average of 4 independent runs per task. Aggregated success rates are normalized relative to the ground truth simulator baseline.

| METHOD | SUCCESS RATE↑ | | | | | | AGGREGATE |
|---|---|---|---|---|---|---|---|
| | ROBOSUITE PUSH | OPEN SLIDE | BLUE BUTTON | GREEN BUTTON | RED BUTTON | UPRIGHT BLOCK | |
| ADAWORLD | $63.50 \pm 1.71\%$ | $5.83 \pm 2.85\%$ | $29.17 \pm 2.50\%$ | $10.83 \pm 2.50\%$ | $10.00 \pm 2.36\%$ | $\mathbf{5.00 \pm 0.96}\%$ | $21.54$ |
| *DiLA* | $\mathbf{68.00 \pm 1.41}\%$ | $\mathbf{15.00 \pm 5.00}\%$ | $\mathbf{78.33 \pm 3.73}\%$ | $\mathbf{35.83 \pm 4.93}\%$ | $\mathbf{20.83 \pm 5.95}\%$ | $3.33 \pm 2.72\%$ | $\mathbf{41.44}$ |

video diffusion models used in *AdaWorld* in visual planning tasks. More details of the visual planning protocol are discussed in Appendix D.

## 5. Discussion

In this work, we introduce *DiLA*, which reconciles the trade-off between action abstraction and generation fidelity by using the predictive bottleneck as a driving force for content-structure disentanglement. Our results demonstrate that this mutually reinforcing process allows *DiLA* to learn continuous action manifolds without requiring explicit supervision. This disentangled representation enables robust performance in challenging downstream tasks, including cross-embodiment action transfer and visual planning.

We also observe several interesting phenomena in action transfer tasks. When the target scene does not support the source action, the transferred rollout still attempts to reproduce the source dynamics as faithfully as possible, which can lead to unusual generations. For example, when the action "walk forward" is transferred to a target scene where a wall is already directly ahead, the generated rollout makes the wall progressively appear larger and blurrier, as if the distance to it were decreasing step by step. An even more striking case occurs when a "throwing" motion is transferred to a robot arm that is not holding any object: the generated rollout may treat the gripper itself as the object being "thrown" to preserve the overall source dynamics as much as possible. Furthermore, when source and target scenes are less similar, the entity that carries the motion may also change. For instance, when we transfer camera-motion dynamics from navigation videos to RT-1-style robot scenes, the motion is sometimes realized as arm movement and at other times as a viewpoint change. In all cases, the target rollout tends to leverage the available object layout in the scene to construct a process that reproduces the source dynamics, even if the result occasionally defies physical plausibility. This also explains why latent actions re-inferred from the target rollout can still preserve much of the original source dynamics.

**Limitations & Future Works** First, the inherent abstraction of latent actions inevitably sacrifices fine-grained control precision, which can lead to instability in downstream video-to-control tasks. The fine-grained control entails highly detailed motion-control information, which is inherently difficult to infer from single-view video alone. In this sense, the challenge is not purely caused by the information bottleneck itself: richer signals such as multi-view observations or robot proprioceptive information would likely be much more effective for injecting fine-grained control information into the representation. For control tasks, we view the latent action as analogous to a high-level action in a hierarchical policy. From this perspective, we generally prefer a smaller bottleneck when the goal is stronger abstraction.

Second, the disentanglement achieved by *DiLA* is limited to separating spatial layouts from visual details; it does not yet support the decoupling of multi-object dynamics. Our regularization is intended to suppress stochastic distractors and favor semantically meaningful motion, but it is not a full object-centric decomposition. That said, *DiLA* has the potential to separate primary dynamics from background dynamics. For example, in a carefully designed dataset where the agent motion is reversible but the background effect evolves only forward in time (e.g., a hand pushes a ball forward and then retracts while the ball keeps moving), the inverse-temporal loss (Eq. 4) would encourage the latent action to encode only the reversible agent motion, while the irreversible background dynamics would be absorbed by the content pathway. We view this as a promising direction for future work.

Finally, our approach remains susceptible to autoregressive compounding errors when generating long-horizon videos.

## Impact Statement

This paper presents work whose goal is to advance the field of machine learning. There are many potential societal consequences of our work, none of which we feel must be specifically highlighted here.

## Acknowledgement

This work was supported by the National Natural Science Foundation of China (no. T2421004 to S.W.), the National Key Research and Development Program of China (2024YFF1206500), the Science and Technology Innovation 2030-Brain Science and Brain-inspired Intelligence Project (no. 2021ZD0200204, S.W.).

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

# A. Implementation details

## A.1. Model parameters

*DiLA* contains approximately 123M trainable parameters. We employ the DINOv2 (base model with registers) and the pre-trained ViT-XL from RAE as our frozen encoder and decoder, respectively (approximately 500M frozen parameters). The hyperparameter specifications for the remaining trainable modules are detailed in Table 5.

*Table 5.* **Model parameters**

| COMPONENT/PARAMETER | VALUE |
|---|:---:|
| **Input parameters** | |
| INPUT IMAGE DIMENSIONS | $256 \times 256 \times 3$ |
| DINOv2 EMBEDDINGS DIMENSIONS | 768 |
| FRAME LENGTHS | 16 |
| PATCH NUMBERS | $16 \times 16$ |
| **Spatial-Temporal Transformer** | |
| PER-PATCH HIDDEN DIMENSION | 768 |
| SPATIAL TRANSFORMER DEPTH | 4 |
| TEMPORAL TRANSFORMER DEPTH | 4 |
| HEADS DIMENSIONS | 64 |
| HEADS NUMBERS | 8 |
| **Structure Encoder (MLP)** | |
| PER-PATCH STRUCTURE DIMENSION | 32 |
| DEPTH | 2 |
| **IDM (3D Convolutions)** | |
| LATENT ACTION DIMENSION (GLOBAL) $d_z$ | 256 |
| CONV BLOCKS NUMBERS | 4 |
| KERNEL SIZE | 3 |
| STRIDE | $(1, 2, 2)$ |
| SPATIAL CONVOLUTIONS | $16 \times 16 \to 8 \times 8 \to 4 \times 4 \to 2 \times 2 \to 1 \times 1$ |
| CHANNELS | $32 \to 64 \to 128 \to 256 \to 512$ |
| **FDM (Lightweight Spatial-Temporal Transformer)** | |
| PER-PATCH HIDDEN DIMENSION | 256 |
| DEPTH | 4 |
| HEADS DIMENSIONS | 32 |
| HEADS NUMBERS | 8 |
| **Content Encoder (MLP)** | |
| PER-PATCH CONTENT DIMENSION | 256 |
| DEPTH | 2 |
| **Content Memory (Mamba)** | |
| PER-PATCH CONTENT DIMENSION | 256 |
| DEPTH | 2 |
| STATE DIMENSION | 32 |
| 1D CONVOLUTION KERNEL SIZE | 4 |
| EXPAND RATIO | 2 |
| DROPOUT RATE | 0.0 |
| **Fusion Decoder (Spatial Transformer)** | |
| PER-PATCH HIDDEN DIMENSION | 768 |
| PER-PATCH CONTEXT DIMENSION | 256 |
| SPATIAL DEPTH | 3 |
| HEADS DIMENSIONS | 32 |
| HEADS NUMBERS | 8 |

## A.2. Training hyperparameters

We implement *DiLA* using the PyTorch framework and train it on a compute node equipped with four NVIDIA A100 (80GB) GPUs. Optimization is performed using AdamW with a learning rate of $1 \times 10^{-4}$, $\beta_1 = 0.9$, $\beta_2 = 0.999$, and a weight decay of $10^{-5}$. Input video sequences are resized to a resolution of $256 \times 256$ and temporally cropped to a sequence length of 16 frames, with a global batch size of 32. The training protocol consists of two stages: initially, the model is trained end-to-end under a teacher-forcing regime for 30k iterations (approximately 24 hours); subsequently, we fine-tune the model for an additional 1k iterations using the latent rollout paradigm to minimize autoregressive error accumulation and enhance

temporal stability. The loss balancing coefficients are set to $\lambda_e = 2.0$, $\lambda_s = 0.03$, $\lambda_z = 0.03$, and $\lambda_{\text{reg}} = 0.001$.

## A.3. Sensitivity Analysis

The coefficients used in the paper are not carefully optimized; we initialized them according to the relative importance of the loss terms, and the model already achieved strong performance. To test robustness, we further adjusted these weights and retrained the model. We observed no significant difference in convergence behavior after the first 10k training steps, suggesting that DiLA does not rely on a narrowly tuned set of loss coefficients and remains stable over a practical range of hyperparameter choices. Meanwhile, the threshold masking mainly regularizes the latent action space. Its form differs across ablated variants (e.g., Discrete and Gaussian), yet these variants can still converge, suggesting that the observed disentanglement effect does not depend on one exact regularizer design. So our claim is not that the co-evolution effect is hyperparameter-independent, but that the factorized architecture + predictive bottleneck enables this interaction in practice.

*Table 6. **DiLA** remains stable over a practical range of hyperparameter choices.*

| Hyperparameters | $L_e$ | Inverse-temporal loss $z$ |
|---|---|---|
| $\lambda_e = 2$, $\lambda_s = 0.03$, $\lambda_z = 0.03$, $\lambda_{reg} = 0.001$ | 0.216 | 0.017 |
| $\lambda_e = 1$, $\lambda_s = 0.01$, $\lambda_z = 0.01$, $\lambda_{reg} = 0.001$ | 0.222 | 0.018 |
| $\lambda_e = 2$, $\lambda_s = 0.05$, $\lambda_z = 0.05$, $\lambda_{reg} = 0.01$ | 0.218 | 0.016 |

We also investigate the effect of varying the bottleneck dimension. Intuitively, a smaller bottleneck forces stronger abstraction and better content invariance, but may slightly decrease the visual accuracy of object interaction. A larger bottleneck preserves more detail, but risks reintroducing nuisance appearance information and weakening disentanglement.

*Table 7. **The effect of varying the bottleneck dimension.***

| Method | Rollout↓ | Cycle Transfer↓ | MSE↓ |
|---|---|---|---|
| Per-patch $d_s = 8$, $d_z = 128$ | 0.418 | 0.446 | 0.269 |
| Per-patch $d_s = 32$, $d_z = 256$ | 0.356 | 0.360 | 0.216 |

In our experiments, a fixed bottleneck setting $d_z = 256$ already works well across synthetic transformations, navigation, robotics video generation, and downstream VP2 planning, suggesting that this is not brittle in practice.

## A.4. Ablation Models

***DiLA* w/o IDM + FDM.**  To verify the necessity of latent action predictive dynamics, we remove the LAM components (both the IDM and FDM). In this variant, the model degrades to a standard video autoencoder where the structure is not predicted from the past but is merely inferred directly from the current frame without any temporal dynamics constraint. This baseline tests whether the predictive bottleneck of the IDM-FDM module is truly the driving force for disentanglement.

***DiLA* w/o content pathway.**  To assess the critical role of our dual-pathway architecture, we ablate the explicit content stream (the Mamba-based memory module). In this monolithic configuration, the model is forced to encode all visual information—comprising both dynamic structure and static appearance—into a single latent representation processed exclusively by the structure pathway. To ensure a fair comparison regarding model capacity, we increase the channel dimension of the per-patch structure embeddings from 32 to 128, yielding a total embedding volume of $16 \times 16 \times 128$. However, we maintain the latent action dimension at $d_z = 256$. This strict bottleneck allows us to evaluate whether a single-stream architecture can resolve the "LAM trade-off" or if it inevitably succumbs to entangled representations and degraded generation fidelity.

***DiLA* w/ discrete $z$.**  We investigate the influence of the latent action space by replacing our continuous bottleneck with a discrete Vector Quantization (VQ) mechanism. Specifically, we adopt the Noise-Substitution VQ (NSVQ) (Vali & Bäckström, 2022) formulation following LAPA (Ye et al., 2024). To maintain an information capacity comparable to our continuous baseline, we configure the VQ layer with a codebook size of 8 and a quantized embedding dimension of 32. With a patch size of 4 (resulting in a $4 \times 4$ token grid), this yields a total flattened latent dimension of $d_z = 4 \times 4 \times 32 = 512$.

This configuration ensures a fair comparison of representational bandwidth, allowing us to isolate the specific effects of discretization on latent actions and video generation quality.

***DiLA* w/ Gaussian $z$.**    To evaluate the efficacy of our geometric regularization strategy (symmetry, norm, and variance), we replace our deterministic constraints with a standard stochastic variational bottleneck. Specifically, we adopt the $\beta$-VAE formulation from Gao et al. (2025), imposing a Kullback-Leibler (KL) divergence penalty towards a standard Gaussian prior (Kingma & Welling, 2013). We configure the regularization weight $\beta = 10^{-4}$ and maintain the latent action dimension at $d_z = 256$. This baseline serves to determine whether standard probabilistic priors are sufficient to structure the latent action manifold, or if our explicit geometric constraints are necessary for optimal performance.

### A.5. Baselines parameters comparison

We include several baselines in the comparison of generation quality, and here we provide the parameter size of each latent action model in Table 8.

*Table 8.* **Model parameters across baselines.**

| Parameters | *DiLA* | LAPA | MOTO | ADAWORLD(LAM) | ADAWORLD | VILLA-X |
|---|---|---|---|---|---|---|
| **Trainable** | 123M | 344M | 440M | 500M | 1.5B | 239M |
| **Frozen** | 500M | - | - | - | - | - |

## B. Latent action analysis details

### B.1. Latent action analysis of single transformation type

For each transformation type, we sample 4,000 unique objects. These objects are initialized at random locations $(x, y) \in [-10, 10]^2$, orientations in $[-90°, 90°]$, and scales in $[0.6, 0.9]$. To account for a warm-up phase, the object begins moving at step 15, at which point we extract the corresponding latent action.

For each regression task, we trained a three-layer MLP adapted from LAPO (Schmidt & Jiang, 2024), featuring hidden dimensions of $(128, 128)$ and ReLU activation functions. The output dimension is set to 1 for all tasks, with the exception of translation direction, which uses an output dimension of 2 to regress the cosine and sine components. We optimize the network using AdamW with a learning rate of $10^{-3}$, weight decay of $0.01$, $\beta = (0.9, 0.999)$, and $\epsilon = 10^{-8}$. All models are trained for 200 epochs with a batch size of 128 using the MSE loss.

### B.2. Latent action analysis of action composition

For each action setting, we sample 500 unique objects. These objects are initialized using the standard protocol: random locations $(x, y) \in [-10, 10]^2$, orientations in $[-90°, 90°]$, and scales in $[0.6, 0.9]$. Distinct from the single-type transformation experiments, the action parameters here are fixed to discrete values: translations of $\pm 20$ units (along $x$ and $y$ axes), scaling factors of $\{0.7, 1.3\}$, and in-plane rotations of $\pm 0.61$ rad ($\approx \pm 35°$). Latent actions are extracted at step 15, marking the onset of motion after a stationary warm-up phase.

### B.3. Latent action analysis of navigation dataset

For the RECON dataset experiment, we sample 4,000 sequences of length 16. To account for the warm-up phase, we utilize the latent action from step 15 of each sequence for analysis. Since the actions in this dataset are composite (combining arbitrary translation and relative yaw), we visualize UMAP of these latent actions by color-coding them according to their relative yaw component.

For the LoopNav dataset experiment, we sample 1,000 sequences of length 16, similarly extracting the latent action at step 15. We filter out "jump" and "pitch" actions due to their scarcity. The filtered data contains no composite actions, comprising 476 forward movements, 242 left turns, and 159 right turns.

## C. Latent action linear probing details

For each dataset, we sample 800 pairs of latent actions $z$ and ground truth actions $a$ from both DiLA and the ablation models to train the probing networks, reserving an additional 200 pairs for testing. We train each linear probing model for 5,000 epochs using an SGD optimizer with a learning rate of 0.01, a batch size of 32, and the MSE loss.

## D. Visual planning protocol

### D.1. Action adaptation

We adapt the action adaptation strategy proposed in AdaWorld (Gao et al., 2025).For each environment, we first sample 100 trajectories to train a projection MLP that maps ground truth actions to latent actions. This MLP consists of two layers with SiLU activations, where the hidden dimension is equal to the latent action size. We train it for 3,000 epochs using an SGD optimizer (lr=0.01, MSE loss) with a batch size of 10.

Subsequently, we substitute the IDM with this pre-trained MLP to derive latent actions directly from ground truth actions. Both the MLP and the DiLA model are then fine-tuned on environment-specific data: 5,000 trajectories for Robosuite and 35,000 perturbed scripted trajectories for RoboDesk. Fine-tuning is performed for 2,000 steps with a batch size of 16 (comparable to AdaWorld, adjusted for memory constraints). We use the AdamW optimizer with a learning rate of $1 \times 10^{-5}$ and optimize only the $\mathcal{L}_e$ and $\mathcal{L}_s$ components of the DiLA training objective.

### D.2. Planning protocol for VP$^2$

Our model is used for planning following the official protocol of VP$^2$. We consider a planning problem conditioned on a goal observation $o_g$ and a historical context $o_{0:L-1}$ of length $L$. Let $t_0 = L - 1$ denote the current time step. We seek an action sequence $a_{t_0:t_0+H-1} = (a_{t_0}, \ldots, a_{t_0+H-1})$ that drives the trajectory towards $o_g$ over a planning horizon $H$.

We address this planning problem using Model Predictive Control (MPC). At each iteration, we sample $N$ action sequences from a distribution initialized to zero. For each sampled sequence, the pre-trained world model predicts a trajectory and computes the associated cost. The sampling distribution is then updated based on a weighted average of the costs, where lower-cost sequences receive higher weights.

Specifically, we use Model-Predictive Path Integral (MPPI) (Williams et al., 2016) to solve this optimization problem. Our implementation follows Nagabandi et al. (2019) as in VP$^2$. At iteration $i \in \{1, \ldots, I\}$, we sample $N$ candidate action sequences $\{\mu^k_{i,t_0:t_0+H-1}\}^N_{k=1}$, evaluate their costs using the world model over planning horizon $H$, and compute a weighted average to derive the updated control sequence $a_{i,t_0:t_0+H-1}$:

$$
\begin{aligned}
a_{i,t_0:t_0+H-1} &= \sum_{k=1}^{N} w^k_i \cdot \mu^k_{i,t_0:t_0+H-1}, \\
w^k_i &= \frac{\exp\left[-\gamma \cdot C(\mu^k_{i,t_0:t_0+H-1})\right]}{\sum_{j=1}^{N} \exp\left[-\gamma \cdot C(\mu^j_{i,t_0:t_0+H-1})\right]},
\end{aligned}
\tag{5}
$$

where $C(\mu^k_{i,t_0:t_0+H-1})$ denotes the cumulative cost of the $k$-th action sequence from time $t_0$ to $t_0 + H - 1$ at iteration $i$.

The sampled action sequences are generated as follows. We initialize $a_{0,t_0:t_0+H-1} = \mathbf{0}$. At each iteration $i \geq 1$, each candidate sequence is obtained by adding correlated noise to the previous iteration's solution:

$$
\mu^k_{i,t} = a_{i-1,t} + \epsilon^k_{i,t}, \quad \text{for all } t \in \{t_0, \ldots, t_0 + H - 1\},
\tag{6}
$$

where the noise sequence $\{\epsilon^k_{i,t}\}^{t_0+H-1}_{t=t_0}$ is temporally correlated via a momentum mechanism:

$$
\begin{aligned}
\epsilon^k_{i,t_0} &\sim \mathcal{N}(0, \sigma^2_{t_0}\mathbf{I}), \\
\epsilon^k_{i,t} &= \beta \cdot \epsilon^k_{i,t-1} + (1-\beta) \cdot \tilde{\epsilon}^k_{i,t}, \quad t \in \{t_0 + 1, \ldots, t_0 + H - 1\},
\end{aligned}
\tag{7}
$$

where $\tilde{\epsilon}^k_{i,t} \sim \mathcal{N}(0, \sigma^2_{i,t}\mathbf{I})$ is independent Gaussian noise with variance $\sigma^2_{i,t}$, and $\beta \in [0, 1]$ is a momentum parameter that controls temporal smoothness. Each sampled action sequence $\mu^k_{i,t_0:t_0+H-1}$ is transformed into a latent action sequence

$z^k_{i,t_0:t_0+H-1}$ via an action encoder MLP, and then fed into the world model together with the encoded historical context $s_{t_0}$ for trajectory generation and cost computation.

We employ the same cost functions as in $VP^2$. Let $\hat{o}_{t+1}$ denote the observation predicted by the world model conditioned on the state-action pair $(s_t, z_t)$, where $z_t$ is the latent action. For Robosuite tasks, the cost is:

$$C(\mu^k_{i,t_0:t_0+H-1}) = \sum_{t=t_0}^{t_0+H-1} \|\hat{o}_{t+1} - o_g\|_2^2. \tag{8}$$

For RoboDesk tasks, let $D$ denote a deep convolutional classifier pre-trained to predict task success. The cost is:

$$C(\mu^k_{i,t_0:t_0+H-1}) = \sum_{t=t_0}^{t_0+H-1} \left( w_p \cdot \|\hat{o}_{t+1} - o_g\|_2^2 + w_D \cdot D(\hat{o}_{t+1}) \right), \tag{9}$$

where we set $w_p = 0.5$ and $w_D = 10$ following $VP^2$. The classifier weights are provided by the benchmark.

In our experiments, we set the number of iterations $I = 15$, context length $L = 2$, planning horizon $H = 10$, momentum $\beta = 0.5$, and temperature $\gamma = 0.05$. For the number of samples, we use $N = 800$ for the *open slide* and *open drawer* tasks, and $N = 200$ for all other tasks.

To evaluate planning performance, we execute each candidate trajectory in the ground truth environment at each iteration and compute the deviation between the goal $o_g$ and the final state as the error. The trajectory with the minimum error across all iterations is selected, and its success rate is reported. Specifically, for Robosuite tabletop pushing tasks, an error below 0.05 is considered a success.

# E. Dataset details

**Something-Something v2.**  Something-Something v2 (Goyal et al., 2017) is a large-scale video dataset that contains 220,847 video clips, focused on human-object interactions that require temporal reasoning to distinguish. Unlike standard action recognition datasets where background context gives away the class, SSv2 focuses on the *motion* of the interaction (e.g., "Pushing something from left to right"). To ensure high-quality training for latent action learning, we applied a filtering strategy (Chen et al., 2025a) to remove static clips or rapid camera motions, resulting in a curated subset that emphasizes clear, distinct physical manipulations.

**RT-1.**  RT-1 (Brohan et al., 2022) is a large-scale, real-world robotics dataset collected using mobile manipulators across diverse office and kitchen environments. It contains over 130k episodes covering a wide range of tasks, including picking, placing, and drawer opening.

**LoopNav.**  LoopNav (Lian et al., 2025) is a benchmark designed for evaluating memory and spatial reasoning within the 3D Minecraft environment. A defining characteristic of this dataset is its discrete, step-by-step control interface. Each time step corresponds to a single, atomic action, ensuring precise alignment between visual changes and action inputs. The action space consists of fundamental navigation commands, including: `forward`, `jump`, ($\Delta$yaw, $\Delta$pitch).

**RECON.**  RECON (Shah et al., 2021) focuses on autonomous ground navigation in unstructured outdoor environments (e.g., grassy fields, gravel, and hills). Unlike the discrete actions in LoopNav, RECON features continuous and compound actions that reflect real-world vehicle dynamics. Specifically, each action is parameterized as a 4-dimensional vector $(\Delta x, \Delta y, \Delta\text{yaw}, \Delta\text{pitch})$, representing the incremental updates to the robot's absolute pose. The dataset includes complex maneuvering behaviors where steering and throttle are coupled, such as sharp left turn, gradual right curve, and varied speed adjustments to navigate traversability constraints. This diversity allows us to test the model's capability to capture continuous motion manifolds.

***OmniObject3D.***  We construct a synthetic dataset featuring 3D rotation, scaling, in-plane rotation, and translation by rendering high-quality scanned meshes from OmniObject3D (Wu et al., 2023b) using Blender. Our dataset comprises 5,911 objects across 216 everyday categories. To enable 3D rotation, each object is initialized at $0°$ and rotated $360°$ around the vertical axis in $5°$ increments, yielding 72 rendered views per object. Additionally, we save the segmentation mask for

each view, which facilitates the synthesis of scaling and translation actions via 2D transformations. We utilize all raw scans provided on the official website; consequently, the number of categories and objects may differ slightly from those reported in the original OmniObject3D paper. The rendering pipeline is adapted from the implementation provided by (Deitke et al., 2023).

## F. Additional visualization

We provide additional visualizations to better understand the generation quality and action transfer capability of *DiLA*.

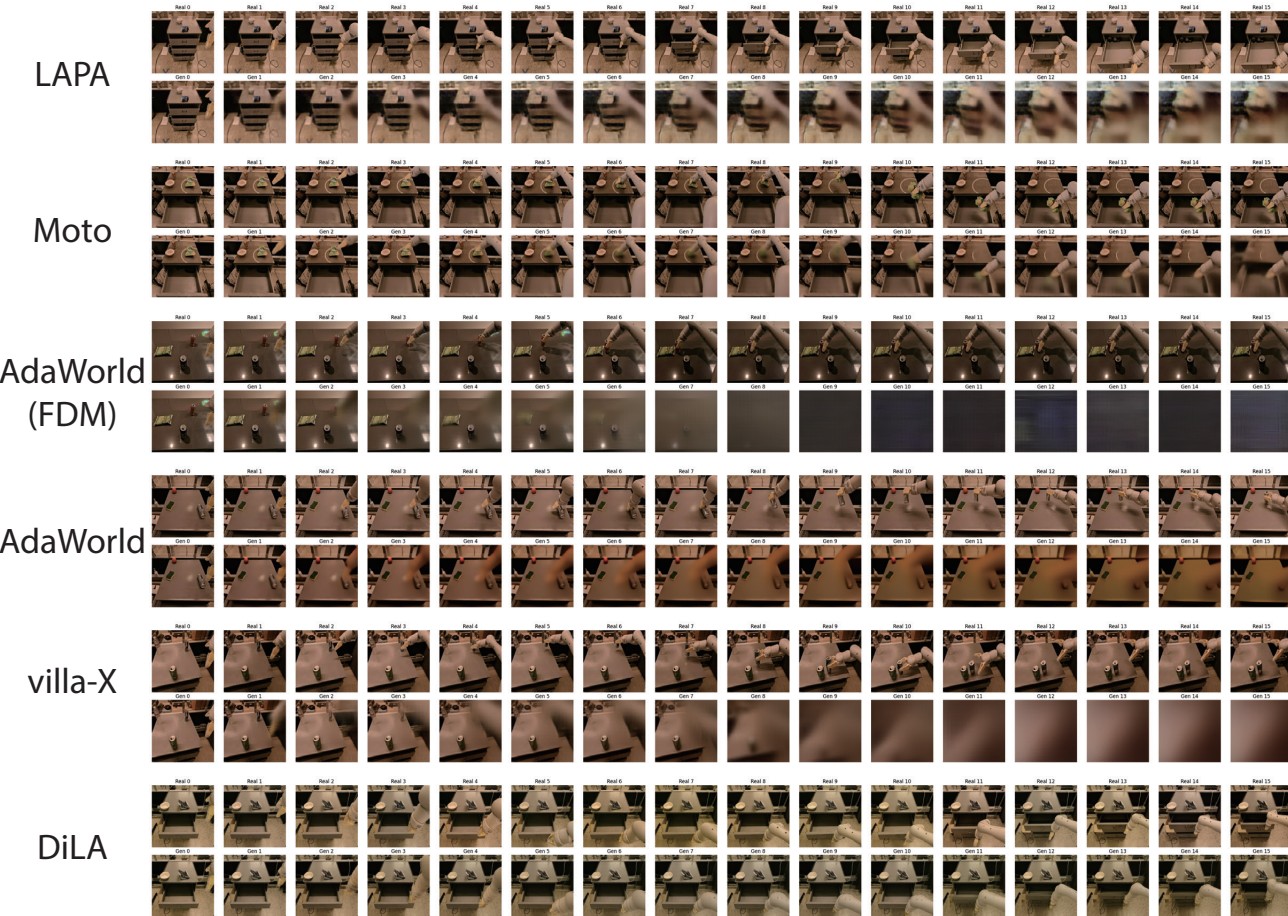

*Figure 7.* **Rollouts visualization of baselines on RT-1 dataset.**

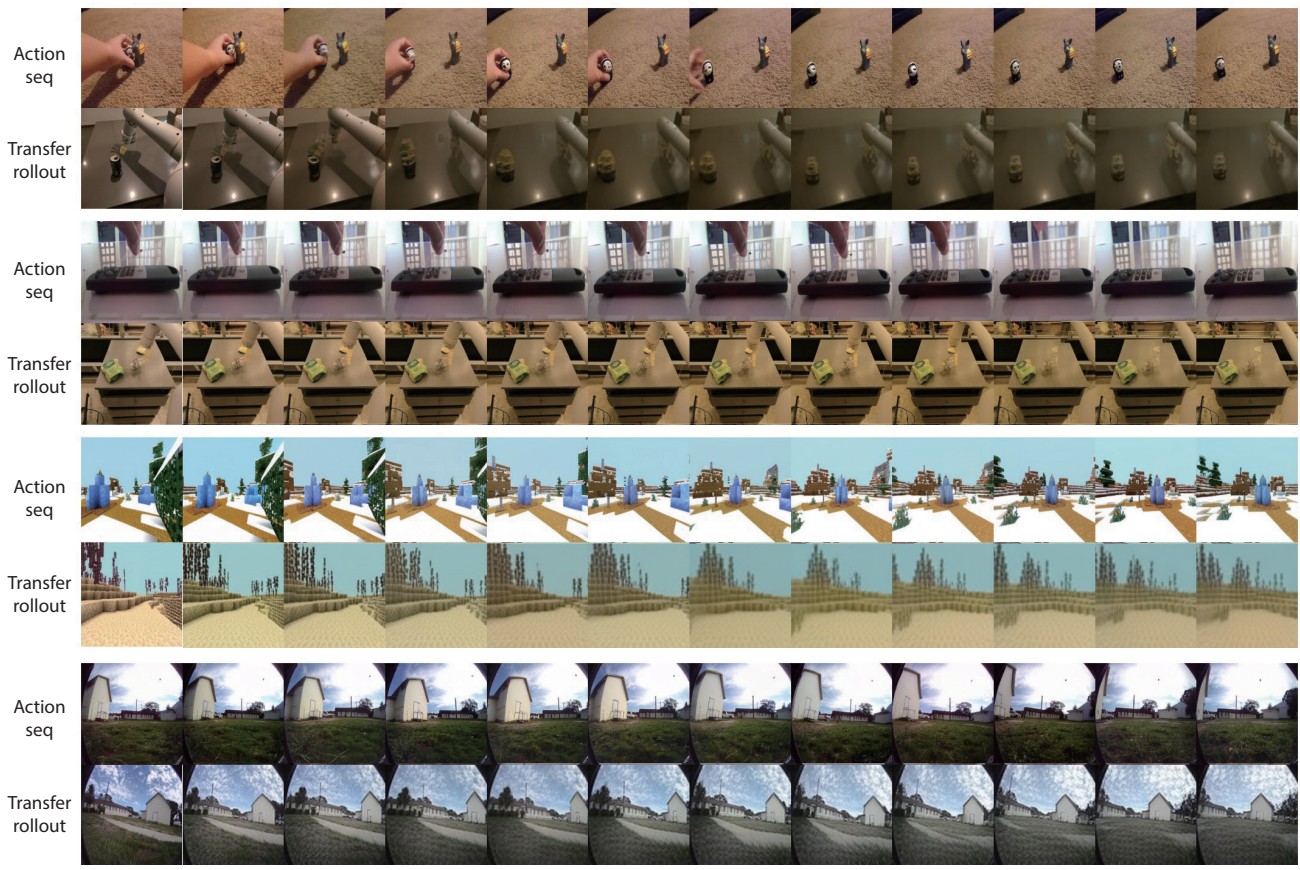

*Figure 8.* **Latent action transfer visualization.**

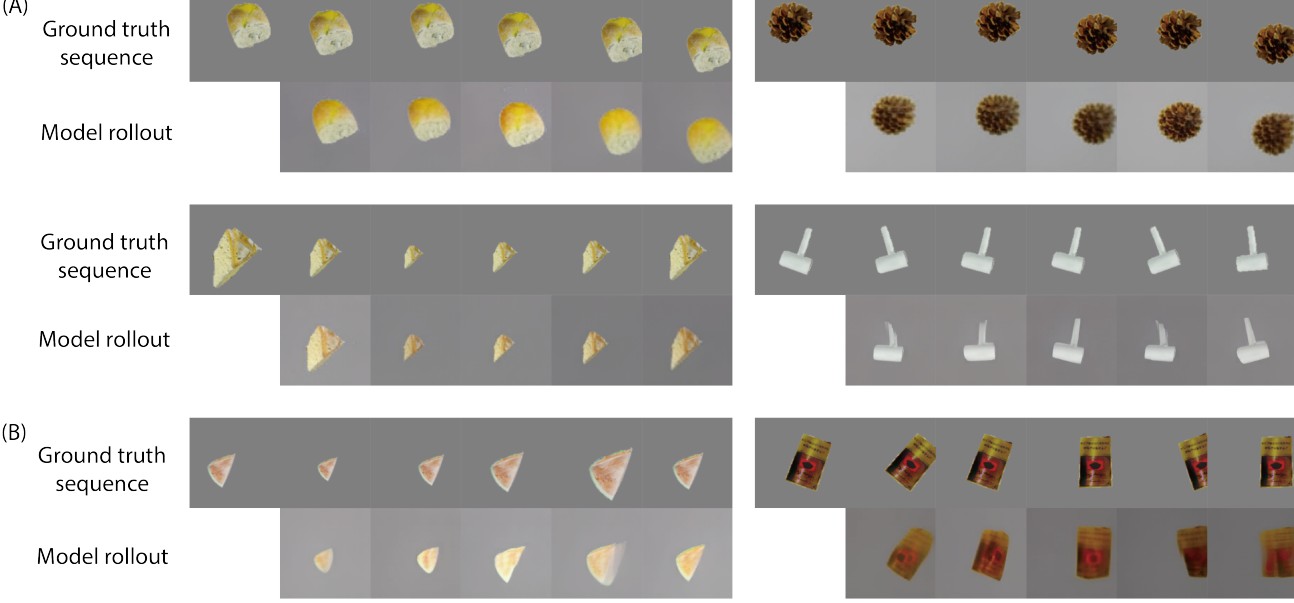

*Figure 9.* **Rollouts visualization on *OmniObject3D* dataset.** (A) Single-type transformations including translation (top), scaling (bottom-left), and rotation (bottom-right). (B) Composite tasks: Translation+Scaling (left) and Translation+Rotation (right)

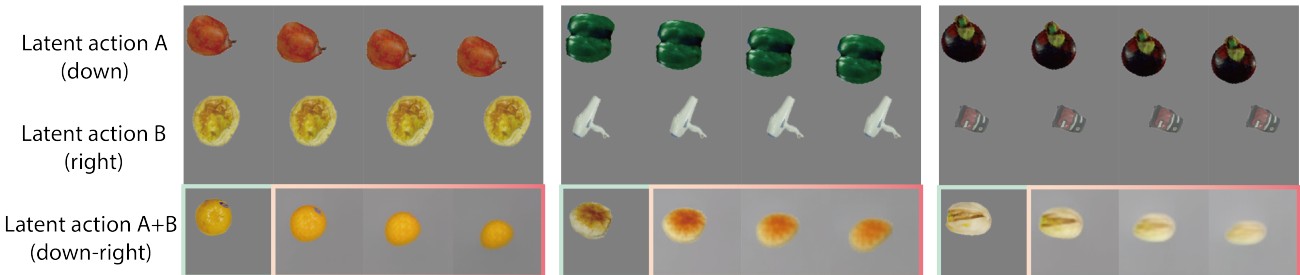

*Figure 10.* **Latent action composition rollout results.**

