# OpenReview forum: "DiLA: Disentangled Latent Action World Models"
_ICML.cc/2026/Conference — ICML 2026 regular_

### Official Review · Reviewer_ZNqy · 2026-03-10

**Soundness:** 3
**Presentation:** 3
**Significance:** 3
**Originality:** 2
**Overall Recommendation:** 4
**Confidence:** 2

**Summary:**

This paper proposes a latent action world model that resolves the trade-off between action abstraction and generation fidelity face by many existing LAMs. The key insight is that latent action learning and disentanglement are co-evolving processes. DiLA processes video through two specialized, parallel pathways including structure pathway and content pathway, enabling separating motion-related and appearance-related information into different embeddings. Structure pathway allows robust action abstraction with only motion-related information. Reported experimental results show that the proposed model outperforms baselines in terms of action accuracy and generation fidelity.

**Compliance With Llm Reviewing Policy:**

Affirmed.

**Key Questions For Authors:**

See above.

**Limitations:**

Yes

**Strengths And Weaknesses:**

**Strength**

1. The paper is well-written and easy to follow.

2. The core insight that the predictive bottleneck of latent action learning can act as a driving force for disentanglement is elegant and effectively addresses the action abstraction and visual generation trade-off.

**Weakness**

1. Villa-X and LAPA comparisons are complicated by different decoder architectures. A more controlled comparison holding the decoder fixed across models would be more convincing.

2. While the "w/o content" ablation is informative, the paper lacks a detailed comparison against a "non-disentangled" model of equivalent total parameter size to verify if the gains are solely due to the architecture or simply increased capacity.

3. The Mamba memory module is central to generation quality (DiLA w/o content degrades substantially), yet the paper provides relatively little analysis of what the content pathway actually learns beyond the motion isolation experiment. What happens with scenes containing significant background dynamics? Does the content pathway silently absorb some motion information in practice?

4. Minor weakness. The objective involves several terms, each accompanied with different weights. Relying on such specific values suggests the model might be sensitive to these settings, making it harder to adapt to entirely new, unseen datasets without extensive tuning.

---

> ### Author Rebuttal · Authors · 2026-03-28
>
> We thank the reviewer for the positive assessment of the paper’s clarity and for recognizing the core insight of our work. We address the concerns below.
>
> W1: Thank you for this insightful observation. We acknowledge that different decoders make our comparisons to LAPA, Moto, and Villa-X less controlled than ideal. These models share a similar LAM structure: a Spatial–Temporal Transformer / ViT-style backbone combined with a bottlenecked vector quantization. Because their latent actions are learned jointly with a **pixel-space reconstruction objective**, simply replacing their decoders with our decoder would substantially change the training setup and, in our view, would no longer constitute a faithful comparison to the original methods.
>
> We fully agree that a controlled comparison is essential, which is why we included the **"DiLA w/o content" ablation**. This variant closely mirrors prior latent action models by removing the disentangled content pathway. Crucially, it isolates the impact of our dual-pathway design by utilizing the same DINOv2 latent prediction objective and fixed RAE decoder as the full DiLA model, ensuring a clean comparison on the decoder side.
>
> To provide an even stronger baseline, this ablation employs our proposed latent action regularization rather than vector quantization (VQ), which yielded poorer generation quality in our experiments. Even with this enhanced setup, the ablated variant performs substantially worse than full DiLA (Table 1, Line 295). This strongly suggests our performance gains stem directly from the proposed disentangled design and its ability to balance abstraction with visual fidelity, rather than merely reflecting a superior decoder choice. Furthermore, the superior performance of this ablated variant compared to the other baselines implies that latent prediction is much better suited for abstract latent action learning than pixel-level prediction.
>
> W2: Thank you for this great suggestion. We would like to clarify that the proposed “w/o content” ablation is **already larger than** full DiLA. To make the comparison conservative in terms of capacity, we enlarge the per-patch structure embedding channels from **32 to 128**, resulting in a 16×16×128 structure representation (Line 353). This gives the w/o content model **646.17M trainable parameters**, compared with **123M for DiLA**. As a result, the gap between the two models cannot be attributed to DiLA having higher capacity. On the contrary, the comparison is favorable to the non-disentangled variant in parameter count, yet it still underperforms DiLA. This result supports the importance of the proposed disentangled design.
>
> W3: Thank you for raising this important question. Regarding scenes with background dynamics: in the current version, if background motion is persistent, it indeed will be encoded in the model. However, such motion will preferentially enter the **structure pathway** rather than the content pathway, because the structure stream is the one directly optimized for temporal prediction through the IDM/FDM. The content memory module is introduced to preserve temporally slow visual details and to support reconstruction of appearance, including regions not consistently observable, rather than to carry the main predictive dynamics. Therefore, **the content pathway will not absorb motion in dynamic scenes.**
>
> That said, DiLA **has the potential to separate primary dynamics from background dynamics.** For example, in a carefully designed dataset where the agent motion is reversible but the background effect evolves only forward in time (e.g., a hand pushes a ball forward and then retracts while the ball keeps moving), the inverse-temporal loss (Eq. 4) would encourage the latent action to encode only the reversible agent motion, while the irreversible background dynamics would be absorbed by the content pathway. We view this as a promising direction for future work.
>
> W4: Thank you for pointing out this ambiguity that we did not make clear. DiLA is **not particularly sensitive** to the loss weights. The coefficients used in the paper are not carefully optimized; we initialized them according to the relative importance of the loss terms, and the model already achieved strong performance. To test robustness, we adjusted these weights and retrained the model. We observed no significant difference in convergence behavior within the first 10k training steps, suggesting that DiLA does not rely on a narrowly tuned set of loss coefficients and remains stable over a practical range of hyperparameter choices.
>
> Table A
> | Hyperparameters | $L_e$ | Inverse-temporal loss $z$  |
> |-|-|-|
> |$\lambda_e=2$, $\lambda_s=0.03$,  $\lambda_z=0.03$, $\lambda_{reg}=0.001$ *| 0.216 | 0.017 |
> |$\lambda_e=1$, $\lambda_s=0.01$,  $\lambda_z=0.01$, $\lambda_{reg}=0.001$| 0.222 | 0.018 |
> |$\lambda_e=2$, $\lambda_s=0.05$,  $\lambda_z=0.05$, $\lambda_{reg}=0.01$|0.218| 0.016 |
>
> > \* indicates the settings used in DiLA.

---

> > ### Author Rebuttal · Reviewer_ZNqy · 2026-04-03
> >
> > The rebuttal addressed most of concerns.

---

> > > ### Author Response · Authors · 2026-04-03
> > >
> > > Thank you very much for your time and for confirming that all of your concerns have been fully addressed. We are pleased that our clarifications resolved your questions, and we sincerely appreciate your constructive feedback and encouraging recognition of the paper’s originality and presentation.

---

### Official Review · Reviewer_riP4 · 2026-03-12

**Soundness:** 4
**Presentation:** 3
**Significance:** 4
**Originality:** 3
**Overall Recommendation:** 5
**Confidence:** 4

**Summary:**

The authors propose a method for learning a latent action space from videos by disentangling it into static and dynamic components, taking care of details such as that even static scene properties may be occluded and hence reveal dynamically. Efficiency is maintained by operating in DINO latent space that can be decoded with a pre-trained decoder. The method is evaluated by transferring actions and appearances, e.g., translating human to robot motion, thereby demonstrating the practical utility of this unsupervised approach.

**Compliance With Llm Reviewing Policy:**

Affirmed.

**Final Justification:**

Thank you for the additional analysis and explanation. My concerns have been fully resolved.

**Key Questions For Authors:**

* what happens if only IDM or FDM are used?
* how sensitive is the model to the hyper parameters, e.g., weights of loss terms. Could this be quantified?
* what would be the effect of varying the bottle neck dimension, would it have to be adjusted for different tasks or datasets (e.g., different size and complexity)? It could be hard to tune?

**Limitations:**

* I would like to see the reliance on the bottle neck to be discussed more explicitly

**Strengths And Weaknesses:**

Significance:
* World models promise to be a playground for AI agents and are therefore widely applicable and even small advanced very valuable

Soundness:
* the method is well motivated and validates the claimed advances with dedicated experiments and visualizations.

Presentation:
* the high level motivation combined with precise equations and descriptive figures makes understanding the method comparably easy

Strength:
* The method outperforms prior work on video generation quality.
* proper ablations are performed, such as validating that generation remains static when only the content latens are used.
* it is also shown that IDM and FDM are crucial.
* The work also scores well (comparable or better) at predictive motion control on the VP2 benchmark, outperforming diffusion models highlights the gains possible by structuring the latent space
* Figure 6 nicely shows the disentangled latent space
* the supplemental document is extensive, providing sufficient detail for re-implementation and insights for future work.

Weaknesses:
* I believe it is difficult to balance the bottleneck size. As noted in the limitation section, it must be small enough leading to information loss for fine-grained control. Hence, the motivated disentanglement and abstraction only works for coarse effects and it remains unclear how to generalize this for details
* Line 207: "patial Transformer" is misleading, see [A]
* the work bears similarity to [B], also using a bottleneck for temporal encoding, and methods cited therein. I suggest discussing the approach more broadly with models that have a similar structure, even if not called (latent action) world model.

[A] Jaderberg, M., Simonyan, K. and Zisserman, A., 2015. Spatial transformer networks. Advances in neural information processing systems, 28.
[B] Rhodin, H., Salzmann, M. and Fua, P., 2018. Unsupervised geometry-aware representation for 3d human pose estimation. In Proceedings of the European conference on computer vision (ECCV) (pp. 750-767).

---

> ### Author Rebuttal · Authors · 2026-03-28
>
> We thank the reviewer for the positive assessment of the paper’s significance, soundness, clarity, and experimental coverage. We are especially encouraged that the reviewer finds the decomposition objective sound, the ablations convincing, and the planning/disentanglement results meaningful. We address the concerns and questions below.
>
> W1: Thank you for raising this insightful question targeting the bottleneck size. A sufficiently constrained bottleneck encourages the model to retain dynamics-relevant structure while discarding appearance-specific nuisance information. In our current design, the latent actions dimension $d_z=256$, which provides a good balance between content invariance and action semantics.
>
> We agree that if the bottleneck becomes too small, some control-relevant detail may be lost. At the same time, we would like to clarify that the fine-grained control referred to in the Limitation entails highly detailed motion-control information, which is inherently difficult to infer from **single-view video** alone. In this sense, the challenge is not purely caused by the information bottleneck itself: richer signals such as **multi-view observations or robot proprioceptive** information would likely be much more effective for injecting fine-grained control information into the representation. For control, we view the latent action as analogous to a high-level action in a hierarchical policy. From this perspective, we generally prefer a smaller bottleneck when the goal is stronger abstraction.
>
> W2: Thank you for your careful reading and for catching this ambiguity. We will correct “spatial Transformer” to “spatial-attention only Transformer” in the revision.
>
> W3: We appreciate this suggestion. We agree that the cited work and related temporal-bottleneck models are closely related to the **content–structure disentanglement** part in our related-work section and should be discussed more broadly, even if they are not framed as world models. We will revise the paper accordingly and clarify that DiLA’s distinction is not merely the use of a bottleneck, but the support of the co-evolution between disentanglement and latent action learning.
>
> Q1: **IDM and FDM are designed as a coupled pair, and both are essential.** With FDM only, next-state prediction still requires an action input, but when IDM is removed, such actions are unavailable, especially in action-free datasets like SSv2. With IDM only, the model can infer latent actions, but without FDM, these actions cannot be used to predict future states, so the predictive information flow is cut off. In this sense, IDM provides the latent action abstraction, while FDM makes that abstraction meaningful by grounding it in future state prediction.
>
> Q2: Thanks for this insightful question. A full quantification of hyperparameter sensitivity would require a large-scale grid search, which is hard due to the rebuttal time limit. However, to address your concern, we did test several alternative settings. Our overall conclusion is that DiLA is **not particularly sensitive** to the loss weights. The coefficients used in the paper are not carefully optimized; we initialized them according to the relative importance of the loss terms, and the model already achieved strong performance. To test robustness, we adjusted these weights and retrained the model. We observed no significant difference in convergence behavior within the **first 10k training steps**, suggesting that DiLA does not rely on a narrowly tuned set of loss coefficients and remains stable over a practical range of hyperparameter choices.
>
> Table A
> | Hyperparameters | $L_e$ | Inverse-temporal loss $z$  |
> |-|-|-|
> |$\lambda_e=2$, $\lambda_s=0.03$,  $\lambda_z=0.03$, $\lambda_{reg}=0.001$ *| 0.216 | 0.017 |
> |$\lambda_e=1$, $\lambda_s=0.01$,  $\lambda_z=0.01$, $\lambda_{reg}=0.001$| 0.222 | 0.018 |
> |$\lambda_e=2$, $\lambda_s=0.05$,  $\lambda_z=0.05$, $\lambda_{reg}=0.01$|0.218| 0.016 |
> >  \* indicates the hyperparameter settings used in DiLA.
>
> Q3: To better address your concern, we have added results varying the bottleneck dimension. Intuitively, a smaller bottleneck forces stronger abstraction and better content invariance, but may slightly decrease the visual accuracy of object interaction. A larger bottleneck preserves more detail, but risks reintroducing nuisance appearance information and weakening disentanglement.
>
> Table B
> | Method| Rollout ↓| Cycle Transfer ↓| MSE ↓|
> |-|-|-|-|
> |Per-patch $d_s=8$, $d_z=128$| 0.418 | 0.446 | 0.269 |
> |Per-patch $d_s=32$, $d_z=256$ *| 0.356 | 0.360 | 0.216 |
> >  \* indicates the settings used in DiLA.
>
> In our experiments, a fixed bottleneck setting $d_z=256$ already works well across synthetic transformations, navigation, robotics video generation, and downstream VP2 planning, suggesting that this is not brittle in practice.

---

> > ### Author Rebuttal · Reviewer_riP4 · 2026-04-03
> >
> > Thank you for the additional analysis and explanation. My concerns have been fully resolved.

---

> > > ### Author Response · Authors · 2026-04-03
> > >
> > > We sincerely thank you for the positive feedback and for increasing the score. We are glad to hear that our additional analysis and explanations fully addressed your concerns. Your constructive comments have significantly helped improve the quality and clarity of our manuscript.

---

### Official Review · Reviewer_Sobw · 2026-03-13

**Soundness:** 3
**Presentation:** 3
**Significance:** 3
**Originality:** 3
**Overall Recommendation:** 5
**Confidence:** 4

**Summary:**

DiLA learns a latent action world model from unlabelled video. The method overcomes the action abstraction vs. generation tradeoff by disentangling content and structure, by capturing actions in the structural pathway and representing the remaining slow-evolving content information in the other. The structure pathway includes the IDM + FDM found in latent action models with an information bottleneck to incentivize the model to distill structural information into this pathway. The content pathway is modeled using a state space model.

In their experiments they show that their content and structure pathways generalize and can be transferred between datasets. The pathways are also important to overcoming the action v. generation tradeoff, where eliminating the content pathway harms generation quality while also reintroducing the tradeoff where the higher action abstraction harms generation. The also perform extensive experiments affirming that no motion is leaked into the content pathway. They also analyze their latent actions by linear readout, and visualizing them via the OmniObject3D dataset. Finally, perform MPC planning on the RoboDesk and Robosuite datasets, outperforming AdaWorld.

**Compliance With Llm Reviewing Policy:**

Affirmed.

**Final Justification:**

They have adressed many of my questions, while there are still some follow-ups I am happy to upgrade my score to a 5.

**Key Questions For Authors:**

- What happens in dynamic scense where other objects are moving, does this information become entangled within the learned latent action space because it involves movement?
- Separately, what if actions performed by an agent are not visible, such as a in an egocentric headcam dataset where parts like hands or torso are not consistently visible?
- Are these actions learned from datasets like RECON and SomethingSomething effective and usable for navigation and manipulation planning tasks?

**Limitations:**

yes

**Strengths And Weaknesses:**

- **Strengths**
    - The decomposition objective is sound and interesting. Removing content that is not beneficial to learning more abstract latent actions seems to be significant and original to the reviewer.
    - The paper and method are generally clearly written and well-explained.
    - Dynamic and static content seems to be successfully partitioned between the pathways, particularly no dynamic movement is present in the content path.
    - The actions learned by the model appear to transfer well between different embodiments of similar morphology (humanoids to humans)
- **Weaknesses**
    - The structure pathway in figure 4 also seems to contain content information and rather the content pathway is more of a “style” pathway. There appears to be a lot of environmental content changes
    - There lacks any quantitative evaluation demonstrating that the action transfer between scenarios and datasets is effective beyond "looking similar" in the presented qualitative examples.

---

> ### Author Rebuttal · Authors · 2026-03-28
>
> Thank you for the thoughtful review and for highlighting the novelty of using disentanglement to improve latent action abstraction, the clarity of the paper, and the strong qualitative transfer results. We address the main concerns and questions below.
>
> W1: Thank you for this feedback, and we would like to clarify that DiLA is not designed to produce a strict split between “content” and “no content.” Instead, the separation is functional: features needed to model latent dynamics under the prediction bottleneck are assigned to the structure pathway.
>
> In locomotion settings, this includes motion-relevant spatial layouts such as pose, shape, and object arrangement, since these are necessary for latent action learning and next-state prediction. By contrast, features that are not directly useful for dynamics prediction, such as texture, color, illumination, and scene style, are encouraged to flow into the content pathway.
>
> So the intended factorization is better understood as **dynamics-relevant spatial structure vs. dynamics-irrelevant visual features**. In this sense, your observation that the content pathway behaves like a style pathway is largely consistent with our design goal. Our claim is therefore not that the structure pathway is entirely free of content information, but that DiLA concentrates motion-relevant attributes in the structure stream while preventing the content stream from carrying the dominant motion signal. We will revise the manuscript to temper our claims accordingly.
>
> W2: Thank you for this valuable point. In fact, we conducted a quantitative evaluation of transfer robustness in Sec. 4.4 (Line 355) through the **action cycle transfer metric**. This metric verifies the preservation of latent action semantics during cross-video transfer by measuring the prediction error after a full cycle of transferring, re-inferring, and reapplying actions. DiLA achieves the strongest cycle-transfer results on SSv2 (Table 2, Line 369). Given SSv2's vast diversity in egocentric motion and scene appearance, these findings offer rigorous quantitative proof of robust transfer across significantly different scenarios, extending well beyond qualitative similarity. To further address your concern, we include **new cycle-transfer results** on a mixture of SSv2 and RT-1, providing broader quantitative evidence that DiLA successfully transfers across entirely distinct datasets and environments.
>
> Table A
> | Method| Rollout ↓| Cycle Transfer ↓|
> |-|-|-|
> |DiLA| 0.2633 | 0.3429 |
> |VQ| 0.3337 | 0.4426 |
> |Gaussian|0.3460|0.4342|
> |w/o content|0.3435|0.4505|
>
> Q1: Indeed, motion from other objects can enter the latent action representation. DiLA disentangles motion-relevant spatial layouts from visual details, but it does not explicitly decouple multi-object dynamics, and we stated this in the Limitations. Our regularization is intended to suppress stochastic distractors and favor semantically meaningful motion, but it is not a full object-centric decomposition. That said, DiLA **has the potential to separate primary dynamics from background dynamics**. For example, in a carefully designed dataset where the agent motion is reversible but the background effect evolves only forward in time (e.g., a hand pushes a ball forward and then retracts while the ball keeps moving), **the inverse-temporal loss** (Eq. 4) would encourage the latent action to encode only the reversible agent motion, while the irreversible background dynamics would be absorbed by the content pathway. We view this as a promising direction for future work.
>
> Q2: DiLA does not require the entire acting body to be visible. The structure pathway operates on temporal changes in the latent structure embedding, so the latent actions contain the information of observable consequences of motion. In SSv2 videos, the hand may be absent in early frames and appear only later. In such cases, the model can still capture the correct motion semantics, but the newly appearing hand in the predicted frame may differ from the ground truth. If the hand has appeared earlier, the content pathway can store its appearance and regenerate it more faithfully. The RT-1 transfer result in Fig. 3 is also an example where the whole limb is not consistently visible. You will find that the transferred robotic arm looks different from the source videos in detail.
>
> Q3: Our evidence suggests yes. The visual planning experiment is intended as a downstream evaluation of whether DiLA learns a stronger latent action representation. For VP2, we train a lightweight action MLP to map executable action labels into DiLA’s latent action space, then fine-tune the DiLA without IDM under the same protocol as prior work. Under this setup, DiLA outperforms AdaWorld on most tasks. In addition, on four OOD robotic benchmarks, DiLA achieves the best linear probing MSE (Table 3), suggesting that its latent actions align better with continuous control signals and are more suitable for downstream control adaptation.

---

> > ### Author Rebuttal · Reviewer_Sobw · 2026-04-03
> >
> > - I like the clarification that it is dynamics-relevant and dynamics-irrelevant features. This makes sense and intuitively supports the decomposition required to improve latent action world modeling.
> > - The cycle transfer metric is relatively compelling and supports the transfer of the learned latent actions. However, this does make me wish to have more clarification on this setting. What is happening if the target setting does not allow for the source action? Ex: source action is walking forward, target setting is standing in front of a wall and cannot walk forward. Additionally, while these latent actions seem to not degrade rollout quality when transferred, reinferred and transferred again, does the necessarily mean the action is “successfully transferred” to the target setting? Is it possible to experience some sort of invariance where the target rollout has nothing to do with the source action but still can be inferred as a sequence of latent actions that replicate the source rollout?
> > - Thank you for the answer to Question 1, I see how this could beneficially affect the learned actions and agree it is an interesting direction for prior work.
> > - Likewise for Q2.
> > - Q3 answers my question as well.
> >
> > I will upgrade my score to a 5.

---

> > > ### Author Response · Authors · 2026-04-04
> > >
> > > Thank you very much for the thoughtful follow-up, and we sincerely appreciate your positive reassessment of our work. We are especially encouraged that the clarification between dynamics-relevant and dynamics-irrelevant features made the decomposition clearer. We also appreciate your deeper questions about the interpretation of the cycle-transfer metric.
> > >
> > > If the target setting does not permit the source action, the transferred rollout will still try to reproduce the source dynamics as faithfully as possible, which can lead to unusual generations. For example, when “walk forward” is transferred to a target scene that is already facing a wall, the generated rollout still makes the wall gradually grow larger and blurrier, as if the distance to the wall were continuing to decrease step by step. An even more interesting case occurs when a “throwing” motion is transferred to a robot arm that is not holding any object: the generated rollout may treat the gripper itself as the object being “thrown,” in order to preserve the overall source dynamics as much as possible. Moreover, when the source and target scenes are less similar, the transferred carrier of the motion may also differ. For instance, when we transfer camera-motion dynamics from navigation videos to RT-1-style robot scenes, the motion is sometimes realized as robot-arm movement, and in other cases as viewpoint change. In all cases, the target rollout tends to use the available object arrangement in the scene to construct a process that reproduces the source dynamics, even if the result is occasionally implausible. This is also why the latent actions re-inferred from the target rollout can still preserve much of the original source dynamics. We will add visualizations and discussion of this phenomenon in the revised version.
> > >
> > > We sincerely thank you again for the careful reading, the constructive suggestions, and the score increase. Your feedback has helped us sharpen both the claims and the presentation of the paper.

---

### Official Review · Reviewer_4BU3 · 2026-03-14

**Soundness:** 3
**Presentation:** 3
**Significance:** 3
**Originality:** 3
**Overall Recommendation:** 4
**Confidence:** 3

**Summary:**

This paper introduces DiLA, a latent action world model that attempts to address the trade-off between action abstraction and generation fidelity by disentangling video representations into a structure pathway and a content pathway. The structure branch learns latent actions from temporal differences and predicts future structural states, while a content memory branch preserves appearance information for reconstruction through a fusion decoder. The paper evaluates DiLA on action transfer, video generation, latent manifold analysis, and visual planning, and reports improvements over several recent latent action modeling baselines.

**Compliance With Llm Reviewing Policy:**

Affirmed.

**Final Justification:**

my concerns are addressed, so the score is upgraded to 4

**Key Questions For Authors:**

See weakness.

**Limitations:**

yes

**Strengths And Weaknesses:**

Strength:
1. The solution is well-motivated, and proposed decomposition is reasonably well engineered.
2. Easy to follow and comprehensive evaluation.

Weakness:
1.The action transfer section is under-quantified.
Section 4.1 and Figure 3 are used to support a major contribution, but the evidence is purely static image sequence. Without a quantitative transfer metric, a user study, or at least a retrieval/consistency measure, it is difficult to assess how robust the claimed cross-embodiment transfer really is. Also,  no video to show the temporal modelling performance.
2. Several key design choices are insufficiently justified in the main paper.
The thresholded masking term in Equation (4) depends on (\tau) and the exact balance of (\lambda_e, \lambda_s, \lambda_z, \lambda_{reg}), both are crucial to the claimed co-evolution of disentanglement and action learning, yet the paper does not analyze this sensitivity. This matters because the claimed mechanism may depend strongly on tuning rather than on the architectural principle itself.
3. The visual planning claim is also only somewhat convincing. Table 4 shows better aggregate performance than AdaWorld on VP², but the setup is no longer pure action-free latent action learning because the paper trains an action MLP and fine-tunes with environment-specific action data. That is acceptable for downstream evaluation, but it weakens any claim that DiLA itself directly enables stronger planning from unlabeled video.

---

> ### Author Rebuttal · Authors · 2026-03-28
>
> Thank you for the positive assessment of the paper’s motivation, engineering quality, clarity, and evaluation breadth. We address the three concerns below.
>
> W1: We appreciate your question on a quantitative evaluation of cross-domain transfer. As demonstrated in our experiments, **the action cycle transfer metric** in Sec. 4.4 (Line 355) is specifically used to assess whether the learned latent actions are abstract enough to support transfer across different contexts. The corresponding results are reported in Table 2 (Line 369). Concretely, latent actions inferred from a source video are transferred to a target video, then re-inferred from the transferred target rollout and applied back to the source. If the transferred action semantics are preserved, the induced increase in prediction error should remain small. This provides a practical way to evaluate cross-embodiment transfer **without requiring ground-truth transferred target videos**. We mainly compared DiLA against ablated variants using this metric on SSv2. Given SSv2's vast diversity in egocentric motion and scene appearance, these findings offer rigorous quantitative proof of robust transfer across different scenarios, extending well beyond qualitative similarity. To further substantiate this, we include new cycle-transfer results on a mixture of SSv2 and RT-1 in Table A, providing broader quantitative evidence that DiLA successfully transfers across distinct datasets and environments. Because the rebuttal format does not allow us to include videos via links, we would like to mention that we provided additional temporal sequence visualizations in Appendix F, which more clearly demonstrate the transfer effect over time.
>
> Table A
> | Method| Rollout ↓| Cycle Transfer ↓|
> |-|-|-|
> |DiLA| 0.2633 | 0.3429 |
> |VQ| 0.3337 | 0.4426 |
> |Gaussian|0.3460|0.4342|
> |w/o content|0.3435|0.4505|
>
> W2: Thanks for this important question! We agree that the balance among $\lambda_e$, $\lambda_s$,  $\lambda_z$, and $\lambda_{reg}$, should be explained more clearly. The coefficients used in the paper are not carefully optimized; we initialized them according to the relative importance of the loss terms, and the model already achieved strong performance. To test robustness, we further adjusted these weights and retrained the model. We observed no significant difference in convergence behavior after the **first 10k training steps**, suggesting that DiLA does not rely on a narrowly tuned set of loss coefficients and remains stable over a practical range of hyperparameter choices. Meanwhile, the threshold masking $\tau$ in $L_{reg}$ mainly regularizes the latent action space. Its form differs across ablated variants (e.g., Discrete $z$ and Gaussian $z$), yet these variants can still converge, suggesting that the observed disentanglement effect does not depend on one exact regularizer design. So our claim is not that the co-evolution effect is hyperparameter-independent, but that the factorized architecture + predictive bottleneck enables this interaction in practice.
>
> Table B
> | Hyperparameters | $L_e$ | Inverse-temporal loss $z$  |
> |-|-|-|
> |$\lambda_e=2$, $\lambda_s=0.03$,  $\lambda_z=0.03$, $\lambda_{reg}=0.001$ *| 0.216 | 0.017 |
> |$\lambda_e=1$, $\lambda_s=0.01$,  $\lambda_z=0.01$, $\lambda_{reg}=0.001$| 0.222 | 0.018 |
> |$\lambda_e=2$, $\lambda_s=0.05$,  $\lambda_z=0.05$, $\lambda_{reg}=0.01$|0.218| 0.016 |
> >  \* indicates the hyperparameter settings used in DiLA.
>
> W3: Thank you for this insightful question. We would like to clarify that latent action learning in DiLA is **fully action-free**. For visual planning, we first directly use the pretrained DiLA to extract latent actions from the downstream robotic datasets. We then train a lightweight action MLP to map ground-truth executable actions to these learned latent actions. After that, we replace the IDM in DiLA with this learned action MLP and fine-tune the rest of the model using environment-specific action data. At this stage, DiLA is adapted into an action-conditioned world model for MPC, but **no latent action learning is performed during fine-tuning**.
>
> In this sense, the planning experiment is not intended to claim that DiLA alone performs end-to-end action-free planning. Rather, it shows that an action-free pretrained latent action model provides a stronger substrate for downstream control adaptation. We will revise the wording accordingly to make this distinction explicit: DiLA is pretrained in a fully action-free manner, then adapted downstream under the same protocol as prior work, where it achieves stronger planning performance than AdaWorld on most tasks.

---

> > ### Author Rebuttal · Reviewer_4BU3 · 2026-04-02
> >
> > my concerns are addressed, so the score is upgraded to 4.

---

> > > ### Author Response · Authors · 2026-04-02
> > >
> > > Dear Reviewer, we sincerely appreciate your positive reassessment of our work and your constructive feedback, which has been instrumental in improving the quality of the manuscript. We remain committed to further strengthening the paper wherever possible.

---

### Decision · Program_Chairs · 2026-04-30

**Decision:**

Accept (regular)

**Comment:**

All reviewers found the proposed method for latent action decomposition to be effective and the results promising, particularly regarding the disentanglement of motion-relevant structure from visual content. The rebuttal successfully addressed initial concerns regarding quantitative transfer metrics, hyperparameter sensitivity, and the role of the content pathway, leading all reviewers to recommend acceptance.

The authors are encouraged to strengthen the final version by incorporating reviewer suggestions, particularly by integrating the new quantitative cycle-transfer results, clarifying the distinction between action-free pretraining and downstream planning adaptation, and ensuring that the functional definitions of the “structure” and “content” pathways are accurately reflected. The authors should also ensure that technical clarifications regarding the bottleneck size and baseline comparisons provided in the rebuttal are included in the final manuscript. There is a clear consensus to accept the paper.